# PandoGen: Generating complete instances of future SARS-CoV-2 sequences using Deep Learning

**Anand Ramachandran**®, **Steven S. Lumetta, Deming Chen**®*

University of Illinois at Urbana-Champaign, Urbana, Illinois, United States of America

* dchen@illinois.edu

**Data Availability Statement:** All data used for the study are available at GISAID's EpiCov repository, as well as from the UniRef database. Accession IDs of GISAID sequences have been deposited in GISAID under the following EPISET_ID:

## Abstract

One of the challenges in a viral pandemic is the emergence of novel variants with different phenotypical characteristics. An ability to forecast future viral individuals at the sequence level enables advance preparation by characterizing the sequences and closing vulnerabilities in current preventative and therapeutic methods. In this article, we explore, in the context of a viral pandemic, the problem of generating complete instances of undiscovered viral protein sequences, which have a high likelihood of being discovered in the future using protein language models. Current approaches to training these models fit model parameters to a known sequence set, which does not suit pandemic forecasting as future sequences differ from known sequences in some respects. To address this, we develop a novel method, called PandoGen, to train protein language models towards the pandemic protein forecasting task. PandoGen combines techniques such as synthetic data generation, conditional sequence generation, and reward-based learning, enabling the model to forecast future sequences, with a high propensity to spread. Applying our method to modeling the SARS-CoV-2 Spike protein sequence, we find empirically that our model forecasts twice as many novel sequences with five times the case counts compared to a model that is 30× larger. Our method forecasts unseen lineages months in advance, whereas models 4× and 30× larger forecast almost no new lineages. When trained on data available up to a month before the onset of important Variants of Concern, our method consistently forecasts sequences belonging to those variants within tight sequence budgets.

## Author summary

Viral protein sequences play a pivotal role in the spread of a pandemic. As the virus evolves, so do the viral proteins, increasing the potency of the virus. Knowledge of future viral protein sequences can be invaluable because it allows us to test the efficacy of preventative and treatment methods against future changes to the virus, and tailor them to such changes early. We attempt to forecast viral proteins ahead of time. Making such predictions is very challenging and complex because the prediction target is a sequence with thousands of positions, and a single mis-predicted sequence position may invalidate the

EPI_SET_230807ta. The UniRef50 database version used in this article is 2019_09. The source code for PandoGen is available at https://github.com/UIUC-ChenLab/PandoGen as open-source software under the MIT license.

**Funding:** This material is based upon work supported by the National Science Foundation under Grant Nos. CNS 1624790, and CNS 1337732. This work also utilizes resources supported by the National Science Foundation's Major Research Instrumentation program, grant #1725729, as well as the University of Illinois at Urbana-Champaign. The funders had no role in study design, data collection and analysis, decision to publish, or preparation of the manuscript.

**Competing interests:** The authors declare that they have no competing interests.

entire prediction. Also, as the virus continues to evolve, the data available to train models becomes obsolete. Addressing these challenges, we create a novel approach to train models of the SARS-CoV-2 Spike protein, that are especially tailored to forecasting future sequences. Models trained using this approach outperform existing approaches in their effectiveness. In addition, our method can train models to forecast important pandemic variants ahead of time.

## 1 Introduction

In the midst of a viral pandemic, many policies and therapeutics are geared towards prevention and advance preparation. However, many such public health and preventative protocols in place are constantly under threat of being upended by evolutionary changes in the virus. In this context, we ask the question—is it possible to computationally generate complete, self-contained, and as-yet undiscovered instances of biological sequences of a virus that is presently causing a viral pandemic? We treat this question within the scope of the SARS-CoV-2 virus, specifically with respect to the Spike protein sequence, which is the initiator of a SARS-CoV-2 infection cycle. An ability to generate viral sequences ahead of time comes with many obvious and powerful benefits. If we know which viral sequences will be encountered in the future, we can study the characteristics of these sequences *in vitro*, and prepare ahead of time, by closing gaps in existing therapeutic interventions or preventative methods.

Recent developments in methods such as Deep Mutational Scanning (DMS) [1] have enabled us to probe the entirety of the Spike protein sequence, and learn properties such as potential for antibody escape, and binding affinity of the protein to the ACE-2 receptor [2], a key step in the host cell entry process for the virus, thus providing effective means to test viral sequences once they are generated.

Coincidentally, in Deep Learning, models used for language modeling, called Large Language Models (LLM) have achieved breakthrough results in generating complete, self-contained, and realistic examples of complex sequence data. LLMs achieve ground-breaking performance in following human instructions faithfully [3, 4] and perform at the human-level in competitive programming [5]. Models based on LLM architectures have recently gained popularity in modeling biological sequences as well [6, 7] and can be useful tools in generating sequences with interesting properties [8–10]. When applied to proteins these models are referred to as Protein Language Models (PLMs) in literature.

PLMs are of two flavors. Masked Language Models (MLM) such as ProtTrans [7] and ESM [11] learn vector representations of proteins at the sequence or residue level by learning to predict randomly masked positions in a known protein sequence. The vector representations learnt by these models are used in downstream tasks for predicting structural and functional characteristics of sequences. The second type of PLMs are autoregressive models such as Prot GPT2 [8], RITA [12], ProGen [10], and ProGen2 [13] which learn to generate protein sequences by sampling one residue at a time, conditioned on all residues generated so far. In addition to inference tasks where properties of a given protein sequence are predicted, these models have also been used to generate artificial protein sequences in an unconstrained manner, or within specific protein families. A hallmark of PLMs is their ability to learn through self-supervision from unlabeled data and make complex predictions, achieving state-of-the-art performance.

Naturally-occurring protein sequences reside in a small subspace of an exponentially large space of sequences defined on the amino acid alphabet. For example, for a sequence length of

1273 (the Spike protein reference sequence length), there are $20^{1273}$ possible amino acid sequences. However, less than a few million unique Spike sequences have been reported so far. In language and protein generation tasks, LLMs and PLMs have shown an ability to generate sensible (for language), plausible (for protein), and sufficiently varied and representative sequences (for both language and protein) from within an exponentially large space of all possible sequences. Hence, PLMs represent a promising method to solve the problem of advance pandemic sequence forecasting end-to-end, which involves recalling a sufficiently large cohort of novel protein sequences with a high chance to be discovered in the future in nature. PLMs' ability to work without complex, expensive annotations or labeling (e.g., sequence alignment or laboratory classification of sequences) represent another significant advantage.

However, the problem of advance generation of the complete Spike protein sequence has not been studied before. While PLMs have been used to label Spike mutations with properties [14, 15] or to generate individual mutations, or sub-sequences within certain regions of the Spike protein [16, 17], they have not been applied to the problem of advance generation of complete Spike protein sequences. Other machine learning methods have been developed to analyze the Spike protein, where they are used to characterize individual Spike mutations or effects of mutations on Spike sub-sequences [18–21]. The goal in these studies is to either identify mutations in existing data that have public health implications, or to identify mutated sub-sequences of the Spike protein that may contribute to the potency of the virus. However, these works are not applied to the problem of generating complete Spike protein instances in advance.

The problem of advance generation of complete Spike protein instances presents some unique and novel challenges that do not exist in other types of generation problems. By advance sequence generation, we imply a specific type of evaluation criterion for the generated sequences: a novel generated sequence must exactly match a sequence that will be discovered in the future, to be considered successful. In addition, the usefulness of generated sequences depends on how infectious the forecasted sequences prove to be. For example, a vast majority of SARS-CoV-2 sequences in the GISAID repository [22] have only been recorded once. These likely represent viral individuals that are not very infectious and hence generating them does not present interesting advance information. We want to generate sequences showing higher propensity to spread, as evidenced by real epidemiological information, rather than through simulated or estimated characteristics using computational or laboratory techniques. That is, while sequence generation happens *in silico*, the evaluation is neither *in silico* or *in vitro*, but against *in vivo* sequences. Let's examine these challenges in greater detail.

First, the requirement of generating complete sequences that exactly match a small discrete set of possibilities is distinct from several popular generation problems in language, vision and bioinformatics. For instance, in language, human evaluation is typically used to determine whether an LLM completed a task correctly, and the results are graded on a scale, with a large number of acceptable responses to a given prompt or prefix [3]. In the case of protein sequence generation, quality is usually determined *in silico* [8, 13] or *in vitro* [9] by measuring a small set of properties of the generated sequences in aggregate, such as sequence similarity, stability, and fold prediction. Prior generative work related to the Spike protein only evaluated individual mutations [16], or generated sub-segments of the Spike protein carrying point mutations using methods that are not scalable to the full sequence generation task [17] (where possible, we have included prior work in our experiments). Thus, the generated output is either a response to a prefix in the case of language (partial generation), not entirely correct or incorrect based on small discrete set membership (language, prior bioinformatics works), or only examines sub-sequences or individual mutations of the virus (partial generation). Unlike these approaches, our problem setting, which requires generating long, self-contained sequences that match a strict solution set is more akin to generating computer code. Like in code, where

a syntax error can cause catastrophic failure, in our case, even a single mispredicted amino acid can result in the answer being invalidated. However, unlike code, we do not have certain tools such as compilers or test-benches that would allow us to filter a large fraction of bad generated samples [5], to obtain a high quality output set.

Second, the problem of generating future sequences is unique to pandemic sequence generation. Traditionally, generative models are trained using a known set of sequences related to a task, during which the models learn the distribution of the training data. Large PLMs do this in two steps: (1) they are pre-trained on a general database of proteins which contains sequences belonging to various different protein families sourced from various different organisms such as those found in the UniProt database [23], and (2) they are finetuned on the specific subfamily of proteins that are desired to be generated [10, 13]. The use-case of the models typically requires them to generate sequences from this distribution of known sequences. In a pandemic situation, there are distinctions between sequences known until a point in time, and new sequences to come, which means that the data distribution changes over time. For example, as the pandemic progresses, sequences accumulate more mutations, and as a result future sequences likely carry a larger number of mutations than the sequences on which models are trained. Sampling from models, which are trained and finetuned to learn the data distribution of known sequences is not suited to our purposes which needs to sample from the future data distribution of unknown sequences, which is different from the known training set. Hence, models need to learn such *arrow of time* effects, and existing pre-training and finetuning methods do not have a mechanism to address this.

Third, generated sequences must be *salient*. In the context of SARS-CoV-2, this means that sampled sequences must have non-trivial case counts in public databases. In data downloaded from GISAID on 2022–12-02, there are 852, 529 unique Spike sequences, out of which only 52, 409 (6.1%) sequences had been reported 10 or more times, and 587, 706 sequences had been reported exactly once. Our goal is that generated sequences from our models should go on to have non-trivial case counts in GISAID, not just that they are simply discovered in the future. Given that the vast majority of available sequences do not fit this criterion, the training scheme needs to use the available data efficiently. It needs to use all sequences to learn the basic structural aspects of the protein to be generated, but at the same time, needs to prioritize characteristics found in a subset of the available data (more infectious sequences) at the time of generating output sequences.

In this article, we propose a method to solve these challenges and train a generative model to forecast future SARS-CoV-2 Spike sequences. As mentioned above, we need the model to learn to generate data according to data distribution that is in the future. However, such data is not available. Hence, we generate training data ourselves from a generative model that is trained following standard practices. This generated training data suffers from the issues discussed before, when taken in aggregate; however, due to the use of sampling, there will be a small subset of the data that has desirable properties, namely unknown sequences with high potential for infectiousness. We mark such subsets in the generated data, and feed it back to the model to learn, where the model learns the characteristics of the type of data that it should not generate, as well as the characteristics of the type of data that it should generate. The cycle of generation → labeling → training is repeated in an iterative manner, helping the model zoom in more and more on the types of sequences that are desired. During this process, guardrails are applied so that the model does not deviate too far from the SARS-CoV-2 Spike sequence space. All the models and methods, including the generative model and the method used to label generated sequences, use only currently known data directly available from public data sources, without additional *in vitro* experiments to characterize synthetic sequences. We will zoom in on a few details below.

During the iterative refinement process described above, we label the generated sequences with two attributes: (1) whether a generated sequence is already known and (2) if it is an unknown sequence, what is its potential for infectiousness. Labeling whether a sequence is known is accomplished by comparing to existing sequences. To label a sequence with its potential for infectiousness, we use a model that we train, called the reward model. The reward model is trained using only data that is already available, that is, the same data that was used to train the generative model. The reward model is trained to determine the winning odds in pairwise games between known sequences. To play a game, all known instances, in a public database, of two sequences are assembled in the game arena, and then an instance is randomly selected from this mix. The sequence whose instance is selected is the winner in the game (Fig 1). It is difficult to define infectiousness potential based purely on sequence occurrence data in public databases, as there may be confounding factors such as maturity of sequence surveillance, number of variants in circulation, disease penetration etc. Through our pairwise game prediction strategy, we avoid the need to learn absolute potentials, and instead help the model learn relative potentials. Also, only *fair* games are played in our training scheme. The

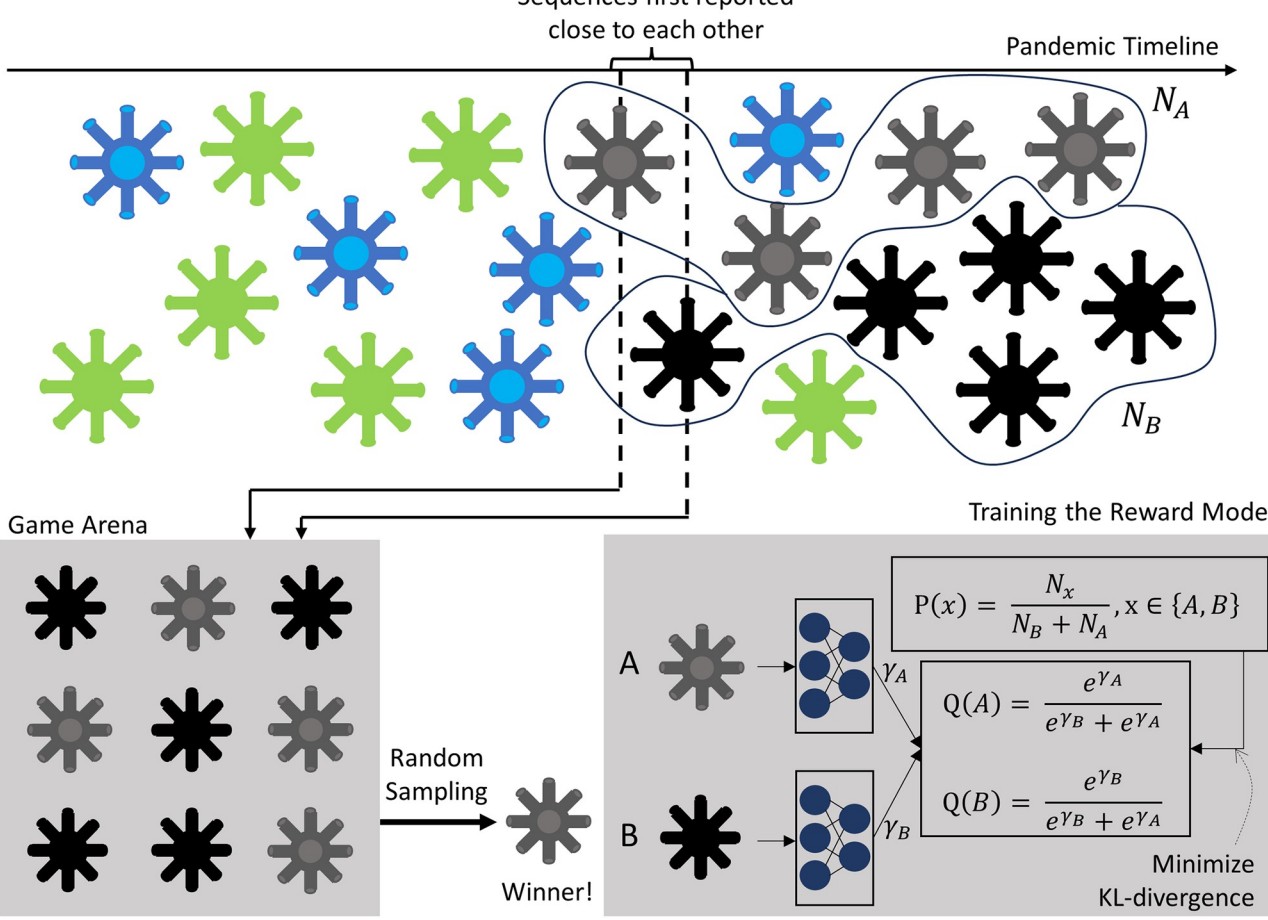

**Fig 1. Training the reward model.** If two sequences are first reported within a short timeframe of each other, a pairwise game can be played between them. To play the game, all instances of the two sequences within the training period are collected into the game arena, and an instance is randomly sampled. The sequence whose instance is sampled is the winner. To train the reward model, we ask the reward model to produce a potential, $\gamma$, for each sequence that is involved in the game. Using these potentials, we calculate the winning probability for each sequence. The ground-truth distribution of wins and losses is calculated from the occurrence counts of the two sequences. The calculated outcome distribution is trained to be similar to the ground-truth distribution.

goal of defining fair games is that the case counts of sequences in public databases are largely reflective of the infectiousness of the sequences. Specifically, for playing a game between two sequences, (1) we need the sequences to be first reported close to each other in time (2) we need sequence count ratios to be similar in two disjoint geographic regions and (3) we require that the sequences have been in circulation for many weeks. These steps are intended to build robustness against variations in sequence case counts due to factors outside of infectiousness. Additional details and limitations of the reward modeling approach are in Section 4.5.

To finetune the generative model using this *in silico* data, we combine two different methods: conditional generation [24] which has been used before to produce sequences of a specific type, and reward-based training [25], which has been used before to improve quality of generated Natural Language sequences, namely to reduce toxicity and improve alignment to human intent. Fig 2 illustrates the procedure. In conditional generation, a generative model is specified a condition, under which to generate a sequence. In our case, we define two conditions.

- Condition 1: to generate known sequences and

- Condition 2: to generate unknown sequences

If an *in silico* training sequence carries a label of "known sequence", during iterative finetuning, the model learns to generate this sequence conditioned on Condition 1. For unknown sequences in the *in silico* dataset, we use the reward model to create floating point scores which are quantized into discrete values. These discrete quanta form a sub-condition within Condition 2. Hence for Condition 2, the model learns to generate sequences based on Condition 2 and a sub-condition that specifies the sequence quality. In practice, the conditions and

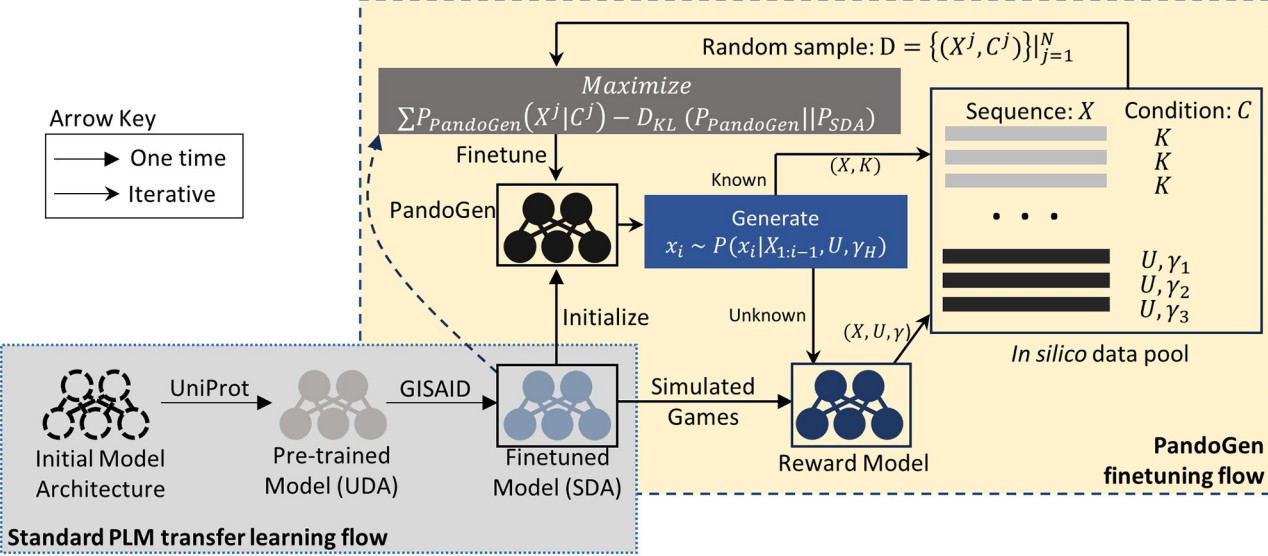

**Fig 2. PandoGen flow.** First, a SARS-CoV-2 deep autoregressive (SDA) model is prepared following standard practices for training PLMs, first pretraining it on UniProt and then finetuning on GISAID sequences. From the SDA model, a reward model is trained to predict winners in pairwise games between known sequences. Next, the SDA model initializes the PandoGen model which is improved in an iterative process. First, the model is used to generate *in silico* sequences, which are then classified as known ($K$) and unknown ($U$) sequences. Unknown sequences are scored using the reward model and scores are quantized ($\gamma$). An *in silico* sequence ($X$), and its labels, ($K$ or ($U$, $\gamma$)) are added to a data pool. Samples from the data pool are used to finetune the PandoGen model while keeping it close to the SDA model. During the generation process PandoGen is conditioned to generate unknown ($U$), highly infectious sequences, which is indicated through the highest reward quantile, $\gamma_H$. $P_{PandoGen}$ is the probability distribution of the PandoGen model, and $P_{SDA}$ is the probability distribution of the SDA model. $D_{KL}$ is the KL-divergence measure between two distributions.

the sub-conditions are flattened into a single generation framework. When the model is used to generate sequences, it is conditioned to generate unknown, high-quality sequences. The combination of conditional generation and reward modeling allow us to inculcate two crucial capabilities that are essential to solving the problem of advance pandemic sequence generation in our model, namely, the ability to learn about the arrow of time by differentiating between known and unknown sequences, and the ability to learn, for unknown sequences, properties related to higher likelihood of being discovered widely in nature.

PandoGen can be applied as a finetuning step on top of existing training and finetuning methods. We applied PandoGen to the problem of modeling the SARS-CoV-2 Spike protein. To evaluate its performance and compare it to other methods, we trained all methods on sequences available until a pre-designated cutoff date, and evaluated sequences generated from the methods on sequences reported in GISAID after that date. In the sequel, we refer to the date on or before the pre-designated cutoff date as the training period. We also trained Pando-Gen on sequences available many days prior to the first reporting of dominant variants of the virus, namely the Delta variant, Omicron BA.5, Omicron BQ.1 and Omicron XBB.1.5 variants, and tested whether sequences generated from PandoGen, within a strict sequence budget, guided by the scale of recent DMS experiments, contain sequences belonging to these variants. The following are a summary of our most significant results.

- We present PandoGen, the first model to forecast complete SARS-CoV-2 Spike protein sequences, generating novel sequences and lineages ahead of time. Our method uses only information available in a public sequence database, and does not need additional *in vitro* experiments or sequence characterizations of any type. We present novel approaches to tailor the model's training process towards the goal of forecasting future SARS-CoV-2 sequences. These approaches differ from standard practices of training Protein Language Models.

- PandoGen presents several new techniques to address the problem of pandemic sequence forecasting. We have tailored and integrated several methods to fit to the problem, namely,

  1. *in silico* training data generation to address the arrow of time effects in known data

  2. conditional generation to encourage model to focus on unknown sequences rather than known sequences

  3. reward-based finetuning to encourage model to produce infectious sequences

- When generating an equal number of sequences, and a similar number of novel sequences, PandoGen forecasts up to four times as many sequences as a model that is 4 times bigger, and two times as many sequences as a model that is 30 times bigger. A novel sequence is a sequence that was never reported in GISAID in the training period. A forecast, is a sequence that is a novel sequence, which will be reported in GISAID after the training period.

- When generating an equal number of sequences and a similar number of novel sequences, sequences forecasted by PandoGen account for 10 times as many cases in GISAID as sequences from a model that is 4 times bigger. The case count ratio is 5 compared to a model that is thirty times bigger.

- PandoGen forecasts many novel lineages, whereas competing methods fail to reliably forecast novel lineages of the virus, instead generating sequences from already known lineages. Similar to a novel sequence, a novel lineage is a lineage that was never reported in GISAID in the training period, and a forecasted lineage is a novel lineage that was reported in GISAID after the training period.

- When asked to generate over 16,000 sequences, PandoGen forecasts lineages first reported up to 10 weeks after the training period, accounting for close to 25% of all novel lineages reported in the 10 week period after the training end date.

- When generating an equal number of sequences, and a similar number of novel sequences, PandoGen forecasts close to 7 times the number of salient sequences as competing methods. By a *salient* sequence, we mean a sequence that has at least 10 cases reported in GISAID through its lifetime.

- When trained on sequences only avilable prior to the reporting of certain dominant variants in GISAID, PandoGen is able to generate sequences belonging to those variants consistently within a strict sequence budget. This holds true for the Delta variant (1 month ahead), Omicron BA.5 (10 days ahead), and Omicron BQ.1 (1 month ahead). We also report that PandoGen failed to forecast the Omicron XBB.1.5 variant.

In the next section, we outline our experiments and results in more detail. In the subsequent section, we describe the methodology behind PandoGen in greater detail.

## 2 Results

We present two sets of results: 1) Quantitative comparisons of PandoGen with baseline methods and 2) Testing the efficacy of PandoGen in forecasting important COVID-19 variants.

### 2.1 Quantitative comparisons

**2.1.1 Training models to forecast SARS-CoV-2 sequences.** As mentioned before, the problem of advance generation of complete viral sequences has not been studied before. Existing approaches for finetuning PLMs depend on transfer learning [26], where a PLM is first pre-trained on a large protein dataset, and then it is finetuned on a problem-specific protein dataset. Both pre-training and finetuning try to fit model parameters to the respective datasets by maximizing the likelihood of the dataset under the model. As PLMs are autoregressive models, the likelihood of a dataset, $\mathcal{D}$, under model parameters, $\Theta$, is written as follows where $L(X)$ represents the length of the sequence $X$, and $\Theta$ represents the parameters of the model.

$$\log P(\mathcal{D}; \Theta) = \sum_{X \in \mathcal{D}} \sum_{i=1}^{L(X)} \log P(x_i | X_{1:i-1}; \Theta) \tag{1}$$

Parameters are typically fit to the data during pre-training and finetuning by calculating the log-likelihood over the data in a batched fashion and improving it through gradient descent. To determine the efficacy of current approaches, and to compare them to PandoGen, we train a model from scratch using prevalent practices, and finetune two pre-trained models from literature on SARS-CoV-2 data. For the model that we trained from scratch, we performed pre-training on the UniProt database and finetuning on SARS-CoV-2 data. This allows us to compare PandoGen to an equivalent model that undergoes all training steps in an identical manner except the final PandoGen finetuning step, thus providing the most faithful measure of the improvement produced by the PandoGen finetuning step.

Prot GPT2 is a protein generation model obtained by training a model based on the GPT2 LLM [27] architecture to generate amino acid token sequences. An amino acid token is a *k*-mer from a pre-determined alphabet of *k*-mers of amino acids, produced from an analysis of a protein repository such as UniProt. The *k*-mers are not necessarily of the same length. Prot GPT2 uses the same tokenization algorithm as GPT2. Prot GPT2 has been applied in limited capacity to the Spike protein modeling problem [16], for the purposes of generating individual

mutations in the Receptor Binding Domain (RBD) sub-segment of the Spike protein. In this article, we present the first application of Prot GPT2 as a *de novo* generator of complete Spike protein sequences. Prot GPT2 has close to 800 million trainable parameters. Among other generative protein language models, we examined ProGen [10], RITA [12] and ProGen2 [13]. We selected the largest model variant from among these, which is ProGen2. Model scaling is a typical approach to obtaining higher performance from LLMs and PLMs [12], hence we will be able to pit PandoGen against this approach by doing this. ProGen2 is similarly pretrained as Prot GPT2, and generates protein sequences one amino acid at a time. The model stands at over 6 billion trainable parameters. Pretrained model parameters for Prot GPT2 and ProGen2 are available for public access. Both are architectures based on transformers [28].

A related work, ProtFound [17], uses masked language modeling to generate mutated sub-sequences within the RBD sub-segment of the Spike protein. To do this, positions in the Spike sub-sequence are masked and the model is asked to fill in the masked locations. Masked language models are not scalable generative models because the number of masks that can be applied to a sequence grows combinatorially with sequence length. To generate instances of the RBD subsegment using masked language modeling, ProtFound utilized supercomputing resources. Hence, scaling to the complete SARS-CoV-2 Spike sequence is infeasible. Due to the scalability issues with masked language models in modeling long sequences, we exclude the method from this article.

For PandoGen, we create a decoder-only transformer model. This model is similar to Prot GPT2 and ProGen2 in that it generates sequences element by element, conditioned on already generated elements. The model operates on individual amino acids like ProGen2, and has close to 200 million trainable parameters, which puts it at a quarter of the size of Prot GPT2 and 1/30th the size of ProGen2.

We pretrained this model on UniRef50 sequences released prior to the COVID-19 pandemic [23]. For convenience, we refer to the pre-trained version of this model as the UniRef50 Deep Autoregressive (UDA) model. Prot GPT2, ProGen2 and UDA models are then finetuned on unique SARS-CoV-2 Spike protein sequences from the GISAID database [22], which were reported on or before a pre-designated cut-off date. The period before this cut-off date is designated the training period. The goal is to forecast sequences that occur after the training period. The period after the training period is referred to as the evaluation period. The UDA model finetuned in this way is referred to as the SARS-CoV-2 Deep Autoregressive (SDA) model in the sequel. The SDA model is finally finetuned using the PandoGen flow described in Section 1. This involves first training the reward model using sequences reported in the training period, followed by finetuning the SDA model according to the flow in Fig 2. The finished SDA model is referred to as PandoGen. The reward model is constructed by taking the SDA model and replacing its final layer with a new layer to produce a reward value rather than perform sequence generation. Further details regarding reward modeling are presented in Section 4.5. Details regarding PandoGen finetuning are in Section 4.6.

For Prot GPT2, we trained a second variant where the training set contains all sequences including repetitions for sequences reported multiple times; that is, if GISAID has a sequence reported *n* times, we have *n* instances of that sequence in the training set. This is done to see whether representing sequences in proportion to their occurrence in the training data will help the model learn the population distribution better. The first Prot GPT2 version, mentioned in the previous paragraph, which is trained on the unique set of SARS-CoV-2 sequences, is referred to as "Prot GPT2 unenumerated" and the second Prot GPT2 version, with the repeated sequences in the training set, is referred to as "Prot GPT2 enumerated" in the sequel. Sequences were sampled from each model finetuned on SARS-CoV-2 Spike sequences using multiple sampling configurations. Except for ProGen2, each model and

sampling configuration was run five or six independent times, each time generating 2048 sequences. Since ProGen2 is a much larger model than the other two, the computational requirements of the model are very high. To accommodate this within the computational resources available to us, we ran the ProGen2 model three times for each configuration generating 2048 sequences each time.

We performed all training and quantitative comparisons using GISAID data snapshot taken on 2022–12-02. Since we evaluate models based on sequences generated by the models after the training period, we need to have sufficiently long evaluation period after the training period to capture the model performance accurately so that all true model predictions are captured in the evaluations. To account for this, we make an allowance for approximately 1.5 years for evaluation, and designate the training period as on or before 2021–06-15. As will be seen later, models forecast sequences many months into the future, and this amount of lead time is warranted for accurate evaluations.

**2.1.2 Sampling algorithms and operating points.** The unrestricted sampling process from a deep autoregressive model proceeds as follows

$$\hat{x}_i \sim P(x_i \mid X_{1:i-1}; \Phi) \tag{2}$$

Here, $\Phi$ represents the model parameters. As shown, the $i^{th}$ sequence element, $x_i$, is sampled from the model distribution, which is conditioned on all previous sequence elements sampled from the model. This is iteratively performed until the end of sequence is encountered. On the other extreme, the following greedy search algorithm is also available, which always yields exactly one sequence, representing one of the most restricted methods of sampling from a model.

$$\hat{x}_i = \arg\max_{x_i} P(x_i \mid X_{1:i-1}; \Phi) \tag{3}$$

There are different methods to tune the model's operating point so that it lies somewhere between the unrestricted sampling process and the restricted greedy sampling process. We adopt nucleus sampling [29], a widely popular sampling method used for LLMs.

To explain nucleus sampling, we define a few terms for clarity first. A sequence is an ordered tuple of sequence elements. Each sequence element is a token that is sampled from the set of all possible tokens that can appear in the sequence, called the sequence alphabet. Nucleus sampling filters away noisy cases from the sampling process in Eq 2 by defining a nucleus, which is a subset of the sequence alphabet, and then sampling from within that nucleus at each step. The nucleus at step $i$ is defined as follows.

$$N_i = \arg\min_{\|M\|}\left(\sum_{x \in M} P(x \mid X_{1:i-1}; \Phi) \geq p\right) \tag{4}$$

Here, $M$ is a subset of the sequence alphabet. The nucleus, $N_i$, is the smallest subset of this set for which the sum of the probabilities at timestep $i$ in the sequence generation process is at least $p$, a parameter of nucleus sampling. If multiple subsets of the same cardinality satisfy the condition, the subset, $M^*$, with the maximum sum $\sum_{x \in M^*} P(x \mid X_{1:i-1}; \Phi)$ can be selected to break the tie. The tokens in the sequence alphabet outside $N_i$ are discarded, and the $i^{th}$ sequence element is sampled from within $N_i$ (the probabilities within $N_i$ are renormalized for this purpose). The maximum value of $p$ is 1 where the nucleus is the complete sequence alphabet. As $p$ decreases, fewer elements are kept inside the nucleus, reducing the number of potential outcomes for each sampling step in the sequence generation process. Essentially, this removes lower likelihood, potentially erroneous, elements from consideration at each

sampling step. However, when $p$ is sufficiently small, the nucleus will begin to exclude non-erroneous, high-information cases. So, at sufficiently low values of $p$, the model would fail to output new information. Hence, varying $p$ allows us to explore the trade-off between error control, and novelty of information in the sampled output.

While $p$ gives us a knob to tune the sampling operation, the value of $p$ alone does not convey a meaningful characterization of the operating point of the model in terms of characteristics of its sequence sample. To provide this characterization, we look to quantify the difference between a generated sample and the known sequences of SARS-CoV-2. A higher degree of difference indicates that generated sequences are likely further apart, evolutionarily, from known sequences. It is natural to expect the models to have a higher error rate when the sample difference with respect to the training set is higher, as models would need to predict further into the future in this case. We expect that lower values of $p$ cause the sample difference to be lower and higher values allow generated sequences to deviate further from the training set.

To quantify sample difference from the known SARS-CoV-2 corpus without expensive computational overheads, we formulate a method using $k$-mer matching. This is based on methods used to determine data quality in genome sequencing [30]. First, we collect all unique SARS-CoV-2 Spike sequences in the training period. We obtain all the $k$-mers in these sequences into a reference $k$-mer set ($k = 11$ in our experiments). Next, for each unique sequence in a generated sample, we count the number of $k$-mers in the sequence that are not found in the reference set. Using this, we find the average per-sequence $k$-mer novelty of the sample with respect to the reference $k$-mer set. We designate this quantity as the measure of sample difference with respect to the reference (training) sample, and refer to this simply as $k$-mer distance. Specifically, if $S_i$ is the number of $k$-mers in the $i^{th}$ sequence that are not found in the reference $k$-mer set, the $k$-mer distance for the sample is $\frac{\sum_{i=1}^{N} S_i}{N}$, where $N$ is the number of unique sequences in the sample. Naturally, the higher the $k$-mer difference of a sample, the further it is from the known set of SARS-CoV-2 sequences.

**2.1.3 Model comparisons.** Prot GPT2 unenumerated, Prot GPT2 enumerated, SDA, ProGen2, and PandoGen were run through multiple sampling configurations. In each configuration, sampling was performed multiple times per model generating 2048 sequences each time. One set of 2048 sequences generated this way is referred to as "a sample" below. The global lower bound for $p$ was chosen to be 0.95 because, at 0.95 the models output very few sequences not already present in the training set (e.g., "Prot GPT2 unenumerated" produces an average of only 31 novel out of 2048 generated sequences), and values lower than 0.95 are expected to produce even less novelty in the output, hence being uninteresting operating points.

Output sequence variety will differ among different models for a given value of $p$. So, we removed sampling configurations that produced too few novel sequences ($<$ 20 novel sequences or 1% of sample size). Second, we wanted to make sure competing models produce a similar number of novel sequences as the best PandoGen sampling configurations so that comparisons are fair to the competing methods. All methods except "Prot GPT2 enumerated" produce comparable sequence diversity in their output samples compared to PandoGen, whereas "Prot GPT2 enumerated", by default, produces very few novel sequences. Hence, we applied a technique called temperature shaping [31] to this model to improve its output diversity. Temperature shaping has been widely used in literature [5, 32, 33], for improving output variety in LLMs. We include the temperature-shaped version of the "Prot GPT2 enumerated" results in the main article. Details regarding temperature shaping are in Section 4.7. In the plots below $T0$, $T1$ refer to the temperature-shaped sampling configuration for "Prot GPT2 enumerated".

Since ProGen2 is a much larger model than the others, to fit the runs in a reasonable amount of time within the computational resources available to us, we further removed the

$p$ = 0.95 configuration for ProGen2, because it was determined to produce the least sequence variety out of all the configurations for the model. This determination was based on a separate small-scale experiment for ProGen2. Details are in Section 4.8.

In all cases, the generated sequences represent the complete Spike amino acid sequence. The Spike amino acid reference has a length of 1273, hence models generate sequences of approximately this length (sequence lengths will vary slightly due to indels).

We measured the following characteristics of the generated samples. Some of these terms were briefly introduced in a previous section, but we repeat them here for clarity.

- **#Forecasts**. This is the number of sequences in the model outputs that were not in the training period, but were reported in GISAID after the training period. Sequences in a model's output that were not present in the training period are referred to as novel sequences. Note that not all novel sequences are forecasts, as some novel sequences are never reported in GISAID. These are considered to be erroneous outputs. A forecasted sequence is a biologically significant, as it represents a stable, fit protein sequence that was realized *in vivo*.

- **#Forecasted lineages**. Number of forecasted lineages from the outputs. A forecasted lineage is a Pango lineage [34] that was not reported in the training period. Pango lineages annotate SARS-CoV-2 sequences with lineage labels tracking evolutionary developments in the SARS-CoV-2 virus based on phylogenetic analysis. The scheme is designed to track and identify lineages that contribute to the spread of the disease. Pango lineage designation is a widely accepted way to track the SARS-CoV-2 virus. New Pango lineages indicate potential changes in local and regional epidemiology and represent potential new information about the pandemic. As such, predicting sequences from new lineages ahead of time is a desirable property for models such as those discussed in this article.

- **#Salient forecasts**. This is the number of *salient* sequences that are forecasted by a model. A salient forecast is a forecasted sequence that has at least 10 cases reported in GISAID. Since a forecast is not a sequence enountered in the training period, all of the occurrences of a salient sequence in GISAID are reported after the end of the training period. Salience is used as a measure of the biological significance of a forecasted sequence. A salient sequence is not only a fit, stable sequence which was realized *in vivo*, but it is also a sequence which has a likely higher propensity to spread.

- **Case counts**. This is the total number of times forecasted sequences in a generated sample are eventually reported in GISAID. Sequences reported within the training period are not included in case count calculation. Case counts constitute an important measure because it shows the ability of the models to predict sequences with higher propensity to spread rather than unimportant sequences with very few case counts.

- **Positive Predictive Value (PPV)**. This is the fraction of novel sequences from the model outputs that were eventually reported in GISAID. That is, if the number of novel sequences in a sample is $N_{novel}$ and the number of forecasted sequences from the sample is $N_{forecast}$, the PPV of the sample is $\frac{N_{forecast}}{N_{novel}}$. Note that PPV is measured exclusively in the subset of novel sequences, and not within the full set of sequences generated by a model.

- **$k$-mer distance**. This is the average $k$-mer distance between the generated sample and the training data as described in Section 2.1.2.

Fig 3 summarizes the characteristics of generated samples from all methods. We plot all quantities against the number of novel sequences in generated samples. Among the methods, ProGen2, the largest model by far, outperforms other methods except PandoGen. PandoGen

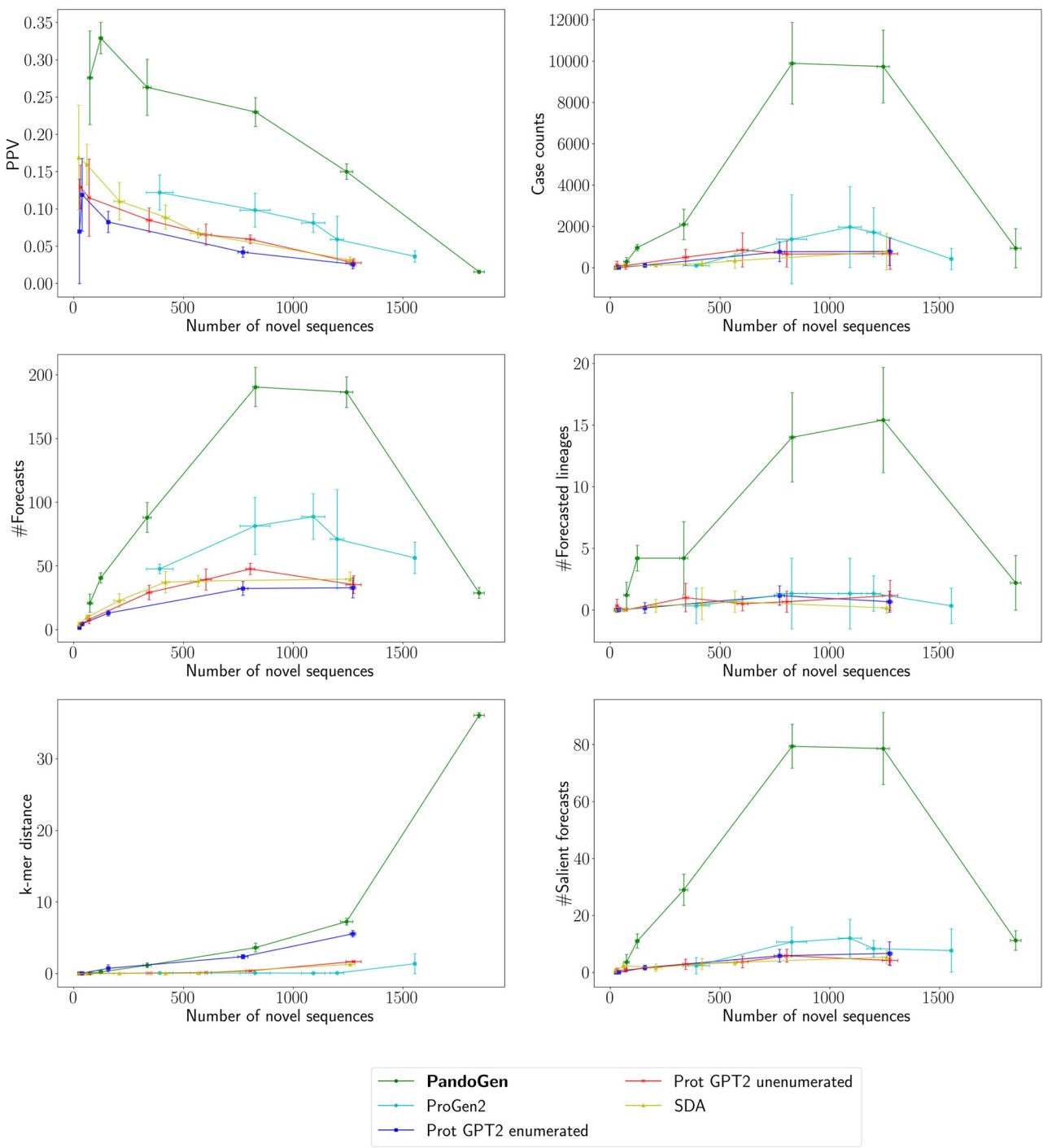

**Fig 3. Sequence sample characteristics from the competing methods.** The *X*-axis of the plots represent the number of novel sequences per sample produced from each method. All experiments are based on sampling 2048 sequences from each model. The spreads represent the 95% Confidence Interval (C.I.) based on multiple sampling runs.

performs better than competing methods across all measures, and in some cases overwhelmingly so. Most notably, the best PandoGen sampling configuration produces sequences with over ten times the case counts as Prot GPT2 and five times the case counts of ProGen2. PandoGen forecasts four times as many sequences as Prot GPT2 and twice as many sequences as

ProGen2. The case counts show a sharper increase than the number of forecasted sequences in both comparisons; this means that PandoGen is able to not only forecast a significantly larger number of Spike protein sequences, but also, these sequences have higher propensity to be infectious compared to other methods. PandoGen also forecasts tens of lineages not reported before. Competing methods notably fail to reliably forecast novel lineages, instead, simply generating sequences from already known lineages. PPV of all methods fall as the methods produce more novel sequences. PandoGen's PPV values are higher than other methods when generating a similar number of novel sequences. Finally, PandoGen produces close to seven times more salient sequences than competing methods. Combined, PandoGen samples have significantly more case counts, salient sequences and novel lineages, indicating that PandoGen samples are qualitatively superior from a pandemic forecasting perspective—outputting biologically novel, and significant sequences compared to other methods.

Overall PandoGen samples tend to have larger $k$-mer distances compared to other methods as well, indicating PandoGen's ability to produce novel information. Related to this, we note that PandoGen's performance at $p = 1$ shows a drop-off. This is because, at $p = 1$, PandoGen samples exhibit a stark increase in the $k$-mer distance. This means the model is trying to predict sequences very different from training sequences, which may be expected to occur further in the future. Making such predictions can be difficult and unreliable. We recommend using $k$-mer distance to filter out operating points which are too ambitious and hence unreliable or error-prone.

Next, we examine sequence generation metrics as a function of sequence rank. This analysis is performed using the same samples as discussed above. As generative sequence models, all methods assign probabilities to generated samples. Sequence ranking is determined based on this probability value. Higher ranking sequences are expected to have better quality, and focusing on higher ranking sequences is a way to get the most reliable data out of a sample set for downstream analyses.

Fig 4 shows the PPV of novel sequences generated from the models, as a function of sequence rank. PPV is higher for all models for higher ranked sequences. ProGen2 again outperforms competing methods except for PandoGen, retaining a flatter PPV curve, meaning that lower ranked sequences from ProGen2 are not as noisy as from competing methods. PandoGen far outperforms all of the competing methods. Notably, the highest ranked novel sequences from PandoGen are at least twice as likely to be sucessful forecasts than other methods with a PPV of over 70%. This lead holds at almost every sequence rank in the top 1000 for the best performing PandoGen configurations ($p \in \{0.995, 0.997\}$).

Fig 5 shows the cumulative number of GISAID cases of the generated novel sequences from the models as a function of sequence rank. Multiple PandoGen configurations dominate the plot. For all methods, the curves flatten out further out from the origin, indicating that the more likely sequences from the methods are also more potent. The curves for competing methods flatten earlier, whereas PandoGen case counts continue to increase for lower-ranked sequences as well.

In Figs 4 and 5, we looked at stats for novel sequence generations from the models. We next look at the fraction of novel, real sequences produced from the models among *all* sequences generated from the models in Fig 6. This fraction is termed "Efficiency" in the figure. The efficiency of all methods is low among the highest likelihood sequences. This is presumably because the highest likelihood sequences are closer to the training data distribution, hence there may be fewer novel sequences among high-likelihood sequences. In the case of PandoGen, the efficiency rises sharply away from the highest ranked sequences, compared to the other methods, which show more modest increases in efficiency. During PandoGen training, we explicitly force the model to learn to generate sequences not in the training set, but we also

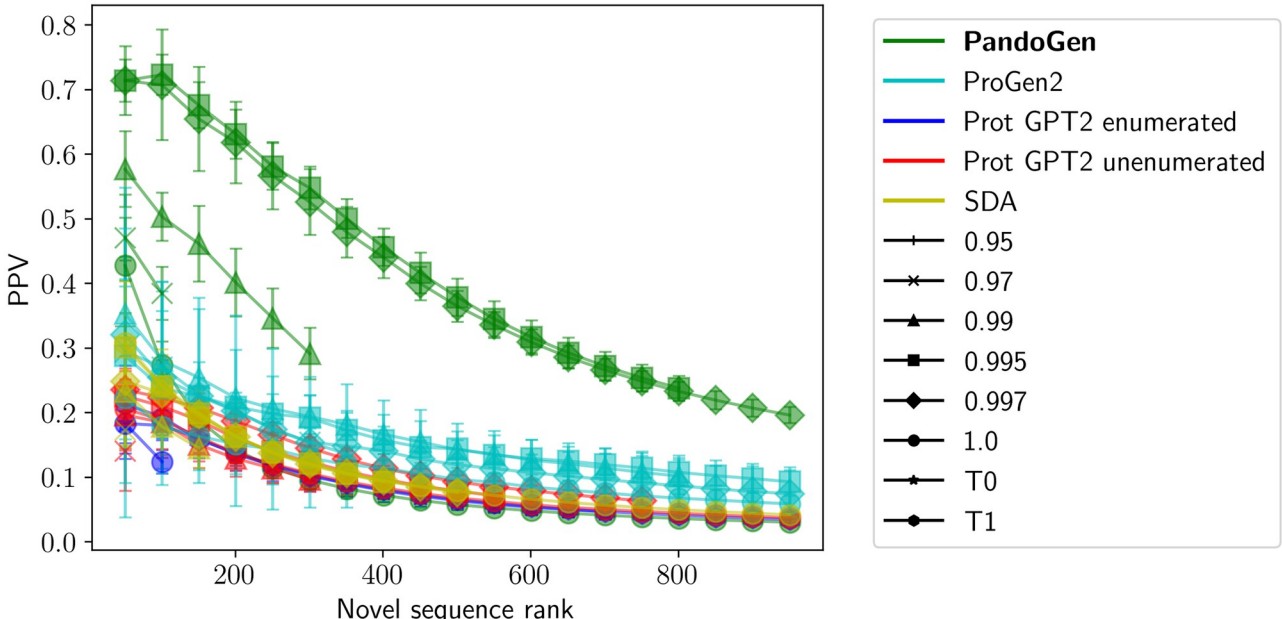

**Fig 4. PPV by sequence rank for the top 1000 novel sequences produced by the different methods.** Error bars represent the 95% confidence intervals for multiple independent sampling runs, each sampling run generating 2048 sequences.

include guardrails to prevent the model from deviating too far from the training distribution to avoid the risk of misrepresenting the viral sequence structure. This leads to dramatically higher sample efficiency except at the highest sequence likelihoods.

Next, we examine the ability of the models to predict into the future. To benchmark this, we plotted the number of sequences, and number of salient sequences forecasted by the models

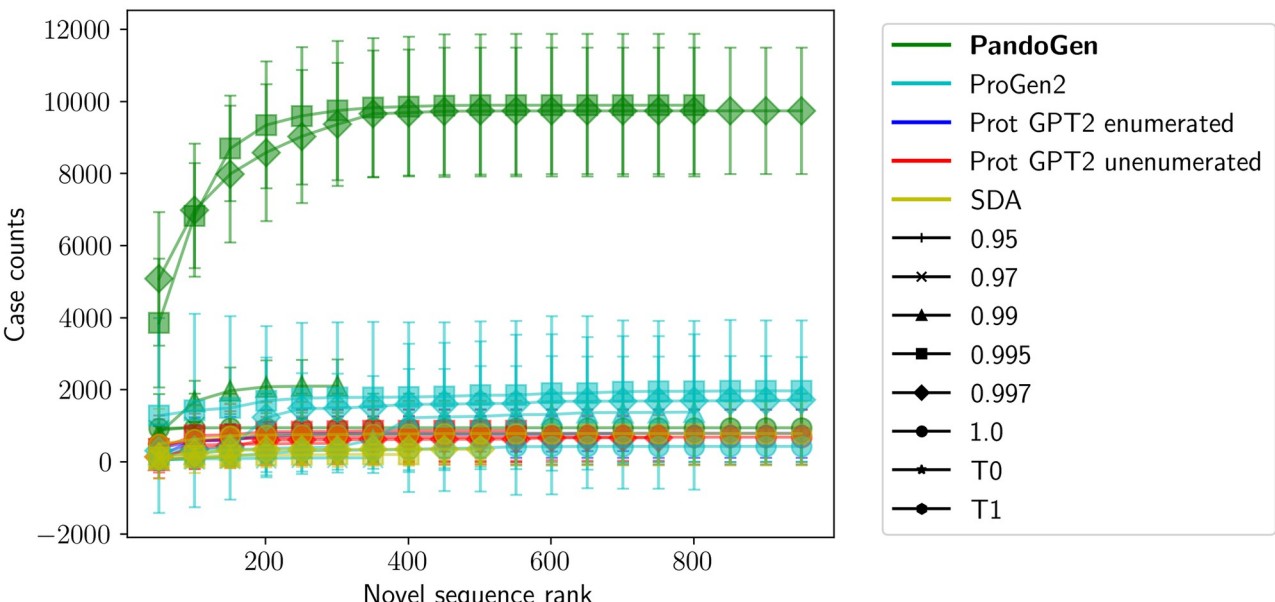

**Fig 5. GISAID case count by sequence rank for the top 1000 novel sequences produced by the different methods.** Error bars represent the 95% confidence intervals for multiple independent sampling runs, each sampling run generating 2048 sequences.

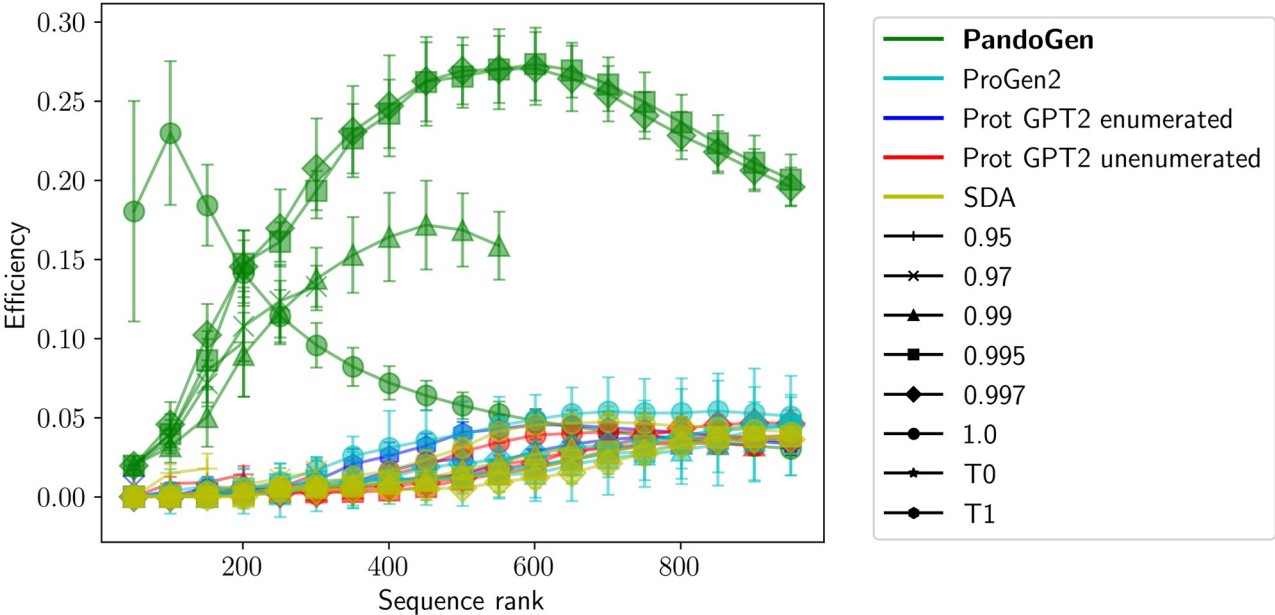

**Fig 6. Fraction of real, novel sequences among all sequences generated from the models by sequence rank.** Error bars represent the 95% confidence intervals for multiple independent sampling runs, each sampling run generating 2048 sequences.

against post-training time in Fig 7. The same samples are used for this analysis as in the prequel. In the figure, a point on the X-axis represents number of months after the end of the training period, say $n$, and Y-axis represents the number of sequences forecasted by the models during the $n^{th}$ month after the training period. For instance, PandoGen forecasts between 40 and 50 sequences in the first month after the end of the training period, out of which between 30 and 40 sequences are salient. From the plots, it is clear that all the models predict several months into the future, with PandoGen maintaining a lead over competitors until month 8 after the training period. Beyond month 10, models make only sporadic forecasts until month 15. PandoGen also holds a significant lead over other methods forecasting salient sequences until month 6 after the training period. No methods produce salient sequences after month 7.

We performed another set of experiments where we scaled model sample size to 16,384 sequences to see how much of the sequence space PandoGen can capture. We observed that when generating 16,384 sequences PandoGen forecasts close to 25% of all novel lineages reported in GISAID the first 10 weeks after the end of the training period. Detailed results are in the Supplementary Document S1 File.

## 2.2 Generating instances of notable variants

Next, the ability of PandoGen to generate important Pandemic variants ahead of time, is examined. Since 2022, various subvariants of the Omicron lineage have practically replaced all other variants of SARS-CoV-2 in circulation and have become the dominant strains of the virus in circulation by far. The BA.5 subvariant of Omicron was the dominant lineage in circulation since mid- to end-2022, which was then replaced by the BQ.1 subvariant [35]. The XBB.1.5 variant is a current variant of concern. Hence these variants are included in the experiments. Prior to the Omicron lineage, the Delta variant (B.1.617.2 and AY.* lineages) caused a large-scale outbreak during the pandemic. To examine PandoGen's performance during the earlier

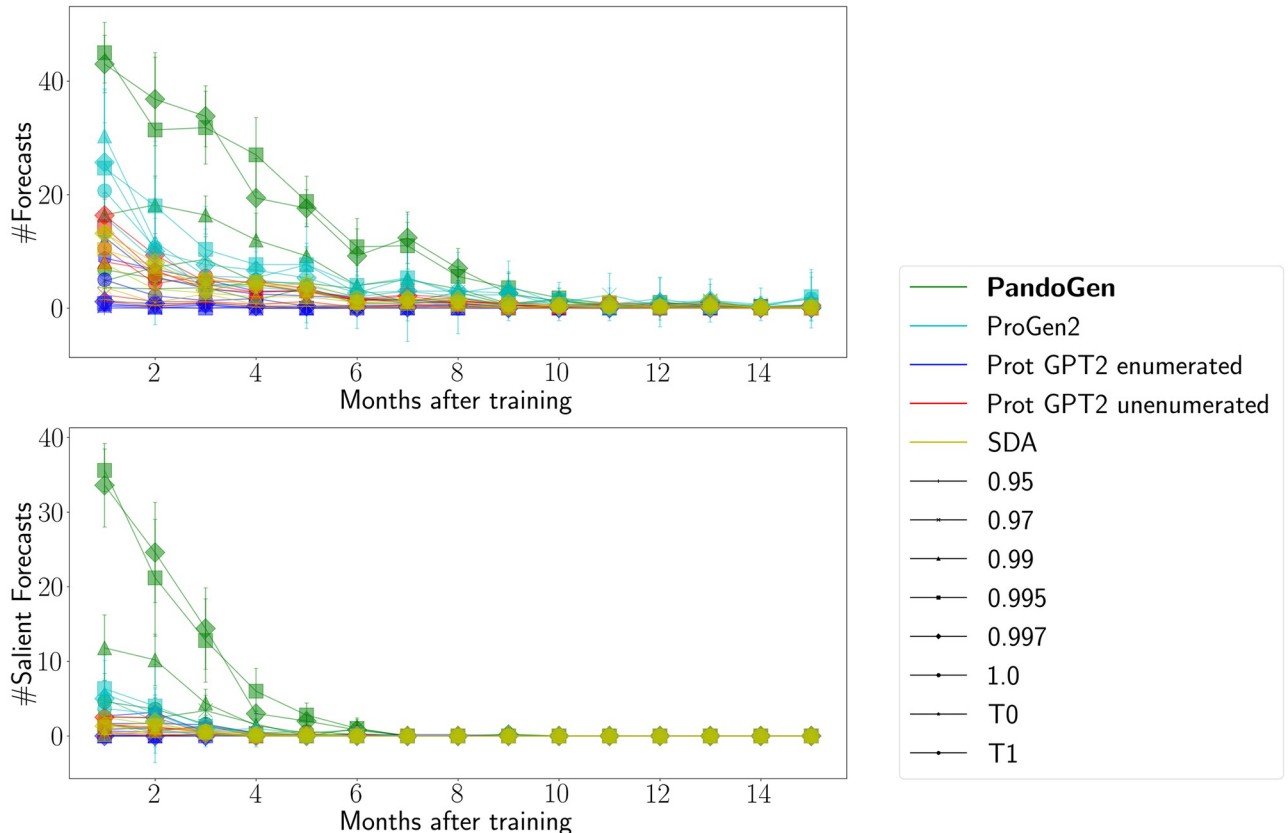

**Fig 7. Forecasted sequences over time.** The X-axis represents months after the training period, and the Y-axis represents the number of sequences forecasted by the models in any given month. For this plot, a month is considered to be a 4-week period.

phases of the pandemic, where fewer sequences are available to train on, Delta was also included in the experiments.

For each variant considered, PandoGen was trained using sequences reported prior to the date on which the first sequence belonging to the given variant was reported in GISAID. To determine the first reported date of a variant, we searched the GISAID database for the corresponding lineage(s) or sub-lineages and took the first Spike protein entry with at least one of the mutations characteristic of the variant. For example, for Delta, we looked for all sequences with lineages matching B.1.617.2 or AY.*, and looked for the first reported sequence with one of the three mutations: T478K, P681R, L452R. We referred prior works for mutations in Delta (B.1.617.2 or AY.*) [36], BA.5 [37], BQ.1 [38], and XBB.1.5 [39] variants. More details are in Table 1. In these experiments, the training cutoff date is between 10 days to a month before the first reported date for these variants in GISAID.

In the intended use-case sequences generated from PandoGen are to be examined in a laboratory setting through methods such as DMS. Hence only a limited sequence budget is assumed. We assume a sequence budget of $\sim 10^5$ sequences, following the budget of full-sequence DMS experiments [1]. To capture a wide range of sequences PandoGen was operated under 11 different nucleus sampling settings with $p \in [0.99, 1.0]$ with a step size of 0.01. For $0.99 \leq p \leq 0.995$ 16, 384 sequences were generated each, and 4096 sequences for the rest of the settings. This is because, at lower values of $p$, the model generates less variety as discussed before. While the total number of sequences generated may exceed the sequence budget, the

**Table 1. Characteristic mutations searched for variants in the experiments.**

| Variant | Characteristic mutations |
|---|---|
| B.1.617.2, AY.* | T478K,P681R,L452R [36] |
| BA.5 | G339, S371F, S373P, S375F, T376A, D405A, R408S, K417N, N440K, L452Q, S477N, T478K, E484A, Q493R, Q498R, N501Y, Y505H [37] |
| BQ.1 | K444T, L452R, N460K, F486V [38] |
| XBB.1.5 | V83A, H146Q, Q183E, V213E, G339H, R346T, L368I, F486S, F490S, G252V, F486P [39] |

number of novel and unique sequences does not, as models generate the same sequence multiple times. Since these are the only sequences that will be tested using DMS, the sequence budget would be honored.

It is also important to see how consistently PandoGen generates a given variant ahead of time. Hence the sampling procedure was repeated many times for each variant. Success is determined by the ability of the model to generate sequences belonging to the targeted lineage or one of its sublineages, as reported in GISAID. In addition to checking the lineage of the generated sequences, we also check whether the sequences contain at least one characteristic mutation belonging to the variant.

The results of the experiments are summarized in Table 2. Except for XBB.1.5, the experiments successfully predicted sequences belonging to the targeted variants ahead of time, and within the requisite sequence budget. For Delta, and BA.5, the predictions are successful in all the experiments. For BQ.1, predictions are successful in 5/6 cases. A higher rate of success may be expected for BQ.1 by increasing the number of sequences generated in this case, as the sequence budget has not been hit in any of these cases.

XBB.1.5 is a sublineage of XBB, a recombinant of two other Omicron lineages. The reward model training was unsuccessful, resulting in a model whose performance was similar to that of random coin-toss for sequences within the training period for XBB.1.5. As a result, the PandoGen pipeline is designated as "failed" for XBB.1.5. This may be because the recombinant parent of XBB.1.5 had been reported only a few weeks before XBB.1.5 itself. In this case, the recombinant breakpoint is located in the Spike sequence's receptor binding domain [40]. The recombinant event does not represent an accumulation of mutations over time such as in the case of most other notable variants of the virus, but the fusion of two other lineages. Hence, from a sequence prediction point of view, this may present some problems for our model, which is based on training data where such events are rare or not notable in effect [40].

**Table 2. Efficacy of generating variants ahead of time using PandoGen (Spreads represent 95% C.I.).**

| VoC | Report date | Training cutoff | #Exp | #Gen | #Success |
|---|---|---|---|---|---|
| B.1.617.2, AY.* | 2021–03–23 | 2021–02–18 | 5 | 11685.6 ± 100.66 | 5/5 |
| BA.5 | 2022–03–15 | 2022–03–05 | 6 | 33599.33 ± 135.77 | 6/6 |
| BQ.1 | 2022–08–08 | 2022–07–08 | 6 | 36401.83 ± 47.42 | 5/6 |
| XBB.1.5 | 2022–10–10 | 2022–09–30 | 0 | N/A | Failed |

Report date: First reported date of variant in GISAID

Training cutoff: Date after which no data is included in training

#Exp: Number of sampling runs

#Gen: Number of generated sequences per run

#Success: Number of sampling runs which forecasted the variant

## 3 Discussion

We presented a novel method to train deep generative models to generate future pandemic sequences with the goal that the generated samples must have a high fraction of real sequences, and that the case count of the generated corpus should be high. We compared models trained through our approach to standard models in the field and found that our method outperforms existing approaches significantly, even when our models are substantially smaller, indicating that alignment of large sequence models to the purposes of generating sequences during a pandemic doesn't automatically happen through transfer learning approaches. Meanwhile, our training pipeline used only information available in GISAID, and did not depend on additional laboratory experiments to perform sequence characterization, making it an attractive approach. We also found that our method is able to predict sequences belonging to important variants before they were available in GISAID.

It is of vital importance that we prepare for the next pandemic. Information-sharing systems such as GISAID were setup very quickly, to combat the COVID-19 pandemic. A system supporting such international data-sharing is vital to analytical efforts such as ours as prediction of future sequences and other important elements of the pandemic can be done more accurately and in a timely manner if the most up to date information is available. We note that there are delays between sequences discovered in the real world versus when they are available in GISAID. But we hope these delays can be significantly reduced with GISAID or another system in case we find ourselves in the unfortunate situation of having to combat another global pandemic. As sequencing infrastructure improves and becomes more commonplace around the world, we have hopes that this will indeed become the case. As such, we hope methods such as ours, will provide valuable tooling supporting pandemic management in the future.

While we presented promising results, we note that there is potential for further research in the future. Being able to predict recombinant lineages is a valuable next step. Recombinant events between two different sublineages typically need co-infection of the same host [41]. Since these events are not as frequent as mutation accumulation events, and since the effect of recombinant events may not be pronounced as frequently, additional considerations will be involved. PandoGen's framework of guiding the generator using the discriminative model can be extended to such cases with some changes. During the training process the generator must generate recombinant candidates and the reward model must rate them as stable, feasible viral sequences. To bootstrap this process and target it towards recombinants, we may need to resort to data augmentation approaches as naturally occurring sequences are not rich in such scenarios. Data augmentation can be envisaged to include the following steps: (1) identification of candidate lineage pairs that may result in recombinants based on sublineages infecting common regions, (2) generation of candidate recombinants by combining the collected pairs, (3) ranking of recombinant candidates through PandoGen models, and through other approaches that can be used to determine protein fitness [42], (4) retraining of the PandoGen reward model to favor candidates ranked high against low-ranked ones and the PandoGen generative model to generate high-ranking candidates, and (5) iterative refinement of the PandoGen generator using the PandoGen reward model. Patterns of prior spread can be used to further improve the process. For example, lineages discovered in a region may show a propensity to spread to other geographic regions due to travel, or nearness, which may be inferred from geographic information present in GISAID. Such patterns can be used to enlarge the set of candidate lineage pairs in step (1) by including nascent lineages in the pairings.

Another potential direction is to integrate PandoGen in a laboratory setting to continually improve the model through incorporation of experimental data. For instance, after PandoGen is trained as in this article, purely using sequence information in GISAID, a second dataset can

be prepared where sequences generated from PandoGen are evaluated in the laboratory setting using DMS to annotate them with multiple properties. Given that these new sequences will have rich annotations that directly reflect the phenotypical characteristics of the virus, this offers an avenue to incorporate more accurate and precise information into a second PandoGen training loop offering more potential improvements. Finally, model sizes [43], training infrastructure [44], and gradient update algorithms [45] are evolving constantly, allowing larger models to be finetuned more efficiently. Such developments will allow bigger models to be incorporated into methods such as ours holding promise for improvements continuing into the future.

# 4 Materials and methods

## 4.1 SARS-CoV-2 sequences

We downloaded the file variant_surveillance.tsv from GISAID, which contains information such as submission date, Pango lineage, list of mutations, geographic location etc. From this file, for each entry, we extracted mutations in the Spike protein sequence, and applied them to the GISAID-designated Spike protein reference to obtain the actual Spike sequences. We collected all sequences except those containing a stop codon mutation, as premature stop codons have a propensity to result in non-functional proteins [46]. The experiments in this article were performed using GISAID variants file downloaded on 2022–12-02. The Accession IDs of sequences in this file have been deposited in GISAID with the following EPISET ID: EPI_SET_230807ta.

We validated the Pango lineage assignments in the GISAID variants file. To do this, we downloaded all SARS-CoV-2 genomic sequences from GISAID, and used the latest Pangolin release, v4.3.1, to assign lineages to these sequences. Out of the GISAID variants file, 613, 093 Accession IDs had lineage assignments that disagreed with our new assignments. Given that the GISAID variants file has 14, 035, 944 entries, this comes to a 95.6% agreement between the GISAID's lineage designation and our independent Pangolin assignment. The disagreeing Accession IDs account for a total of 112, 116 unique Spike sequences, and all Accession IDs in the GISAID variants file account for 852, 529 unique Spike sequences, which means, at the unique sequence level there is an 86.8% agreement between the GISAID assignments and our independent assignments. When looking only at sequences in the training period, the agreement for Accession IDs is 97.8% and the agreement for unique Spike protein sequences is 90.5%. Additional details are in the Supplementary Document S1 File. We used the new lineage assignments to perform all evaluations presented in this article.

For comparing sequences, we allow ambiguous amino acid characters to match with their corresponding disambiguated targets. That is, we consider "B" to be equal to either "D" or "N", "J" to be equal to either "L" or "I", "Z" to be either "Q" or "E", and "X" to be any amino acid. The matching algorithm, in full detail, is given in the Supplementary Document S1 File. Sequences are first checked to see whether they match sequences reported in the training period. Sequences not matching sequences in the training period are designated as novel sequences. Novel sequences are checked among sequences first reported after the training period to determine which novel sequences are real sequences, or successful forecasts.

For reporting the number of generated sequences in Section 2.2, where only the order of magnitude of unique sequences is relevant, we used the computationally inexpensive exact string matching predefined in Python.

## 4.2 Model architecture

We base our models on decoder-only autoregressive models, which consist of stacks of modules, each module consisting of a self-attention layer, a layer-normalization layer, a non-

linearity and dropout layer [28]. The model has eight of these modules, with attention head dimensionality of 128, with 12 attention heads, and a fully connected layer that is three times the attention size. On top of the 8 stacked modules, we have a final Softmax layer that predicts a distribution over the protein sequence elements (amino acids). We call this final layer, the Protein Model head (PM head), for ease of reference later. Overall, there are approximately 192 million parameters in our models (SDA and PandoGen).

Given a training set, each amino acid character in a protein sequence in the training set is converted to an embedding; hence the input protein sequence becomes a sequence of embeddings of the same length. To this sequence, we prepend a special character indicating start of sequence. To this, a second embedding sequence of the same length and dimensionality is added. This second embedding indicates the position of each embedding in the sequence, and is trained along with the rest of the model. These conventions largely follow standard Transformer models [28].

The model predicts the probability of a protein sequence as follows.

$$\log P[X_{1:N}] = \sum_i \log P(x_i \mid X_{1:i-1}; \Phi)$$

(5)

Here, $\Phi$ represents the parameters of the model, and $P(x_i \mid X_{1:i-1}; \Phi)$ is the output of the PM head. During data generation from the model, the generation is conditioned on the start of sequence character, that is, we ask the model to complete the sequence provided only the start of sequence special character.

## 4.3 Pretraining on UniProt dataset

We downloaded the UniProt version 2019−09 which contains protein sequences released before the onset of the COVID-19 pandemic. We first pretrained our 192M parameter model using UniRef50 sequences, split randomly into 35M training sequences and 3.9M validation sequences. Pretraining was done for our models using snippets of UniRef50 sequences of length 255. Where a sequence was shorter than length 255, the complete sequence was used and when sequence was longer, a random substring of length 255 was sliced out. During validation, to keep the validation set deterministic, we used only sequences shorter than length 256. We trained the model for two epochs with a learning rate of $1 \times 10^{-4}$, and a weight decay of $1 \times 10^{-2}$. We employed learning rate warmup for 1024 batches, and thereafter, a linear decay in learning rate was instituted. We used 4 NVIDIA V100 GPUs to pretrain the model using half-precision floating point with a batch-size of 30 per device. Validation loss was calculated once every quarter of an epoch, and the model version with the best validation loss was retained for further finetuning. We call this model, the UniProt Deep autoregressive Model (UDA), for ease of reference.

## 4.4 Finetuning on SARS-CoV-2 Spike protein sequences

To create the training dataset from GISAID, we selected a cutoff date, and sequences on or before this cutoff date were used for training. The goal would be to predict sequences after this cutoff date.

For training deep learning models, typically, the training data is split into a training set, on which the model parameters are fit, and a validation set, on which the models are continually evaluated using the loss function. The model in the training iteration that has the best loss value on the validation set is selected for downstream experiments. This avoids overfitting the models to the training set, which Deep Learning models show a propensity to memorize during the course of training. To determine the training/validation split, we used Pango lineages

reported in GISAID, splitting the lineages into training and validation lineages. Sequences belonging to training lineages were used designated the training set and the remaining sequences were designated as validation sequences. Any Spike sequence occurring in both a training and a validation Pango lineage was moved to the training set. Note that the Pango lineage assignments are themselves not fed to the models. Presumably, other methods of splitting the training data into training and validation set would also work, such as a simple random split, or another method of clustering, such as by dates or using a machine learning method such as K-Means clustering operating on sequence embeddings. The manner of split need not be exact in some sense. All of the generative models described in this article are trained on the same training and validation set split, hence the same data split is applied everywhere. As mentioned before, for evaluations we did not use the lineages reported by GISAID, but rather lineage assignments obtained by us after rerunning Pangolin.

We finetuned the model resulting from Section 4.3 as well as the Prot GPT2 unenumerated model and ProGen2, on this same training/validation split, for 24 epochs, checkpointing the models twice per epoch. All models used a fine-tuning learning rate of $1 \times 10^{-5}$, with a weight decay of $1 \times 10^{-2}$, 1024 learning rate linear warmup steps and linear decay afterwards. All models were trained using 16-bit floating point weights. We evaluated each checkpoint on the validation set, and selected the one with the best loss. We term the finetuned version of our UDA model as the SARS-CoV-2 deep autoregressive model (or SDA model) in the sequel.

In addition, we trained "Prot GPT2 enumerated" where the training sequences are not made unique. So, if a sequence is deposited in GISAID 500 times, the sequence occurs in the training set 500 times as well. The training data size for this case resulted in a similar number of gradient updates for a single epoch as 24 epochs of the data with unique sequences used for finetuning the other models. Hence training in this case was performed for a single epoch only. The training and validation split for the model is the same as that of the other models, just that the sequences may be repeated depending on their GISAID frequencies.

## 4.5 Reward model training

The reward model's goal is to predict potential infectiousness for an input sequence as a scalar value. We would like to use the number of times a Spike protein sequence has been reported in the GISAID database as a signal indicating the infectiousness of the sequence. However, it is not straightforward to use this data due to the following reasons. First, a sequence discovered earlier during the pandemic has had more time to spread, compared to a newer sequence. Second, sequencing resources may have needed time to ramp up around the world. Third, restrictions and interventions against the pandemic have shown variation over time. As a result of these, using the raw GISAID case counts as a label is misleading. For instance, the original strain of the SARS-CoV-2 virus discovered in 2019 has had many years to circulate, whereas a more recently discovered sequence has had very little time, but their case counts may imply that the original strain is more infectious, which is not necessarily true.

We simulate a game between two sequences. The arena for the game between Seq A and Seq B, is the set of all GISAID cases within the training period that belong to Seq A and Seq B. The game is to randomly select an item from this set. Seq A wins if the selected item is Seq A, and Seq B wins otherwise. The probability of Seq A winning a game is $\frac{N_A}{N_A + N_B}$ where $N_A$, $N_B$ are the GISAID case counts of Seq A and Seq B respectively.

We ensure that the games are played only between sequences whose first reported date in GISAID is within a narrow timeframe of each other. Implicit here is the assumption that this ensures that they were under similar constraints throughout their lifetimes within the training period, and that their case counts are sufficiently indicative of their relative infectiousness.

**Table 3. Ratio of case counts of the lower-occurrence sequence to that of the higher-occurrence sequence and allowed tolerance in their weekly discovery dates for them to be allowed to be compared.**

| Case count ratio ($r$) | Discovery week tolerance |
|---|---|
| $0 \leq r < 1e - 5$ | 5 |
| $1e - 5 \leq r < 1e - 4$ | 4 |
| $1e - 4 \leq r < 1e - 3$ | 3 |
| $1e - 3 \leq r < 1e - 2$ | 2 |
| $1e - 2 \leq r < 1e - 1$ | 1 |
| $1e - 1 \leq r < 1$ | 0 |

Sequences can have a small difference between their respective discovery dates (discovery dates are dates on which the sequences are first reported in GISAID), depending on how clear, on average, the winner is, between the two sequences. The largest difference in sequence discovery date in our dataset is 5 weeks, but this requires one sequence to have at least $10^5$ times more cases than the other. As the ratio of case counts between sequences shrinks, the allowed gap between their discovery dates is also tightened. The scheme is detailed in Table 3. As shown, as the case count ratio between two sequences increases, they are allowed to be discovered further apart in time to be inducted into the competition draw.

Another condition to include a pair of sequences in the competition draw is that their relative case counts must be similar in disjoint geographic regions, and the expected winner in the majority of the games between the two sequences must be unchanged between the two regions. That is, if Sequence A and Sequence B are being considered for inclusion in the competition draw, we require that the relative counts of the two sequences within Asia and Europe must be similar to their relative case counts outside of Asia and Europe, within the training period. In addition, if Sequence A leads in Asia and Europe, it should also be the leader outside of Asia and Europe—that is the majority winner in these two disjoint regions must be the same.

Finally, we do not use any sequences first reported in the last 7 weeks of the training period (sequences reported within the 7 week period, which were first reported before that period, are still considered), as the sequences discovered in this time frame may not have had sufficient time to spread.

Examples of considerations when creating the competition draw are illustrated in Fig 8. Consider pairs of sequences which satisfy the criteria that 1) they have similar relative case counts in Eurasia and the rest of the world, 2) the majority winner is the same in Eurasia vs

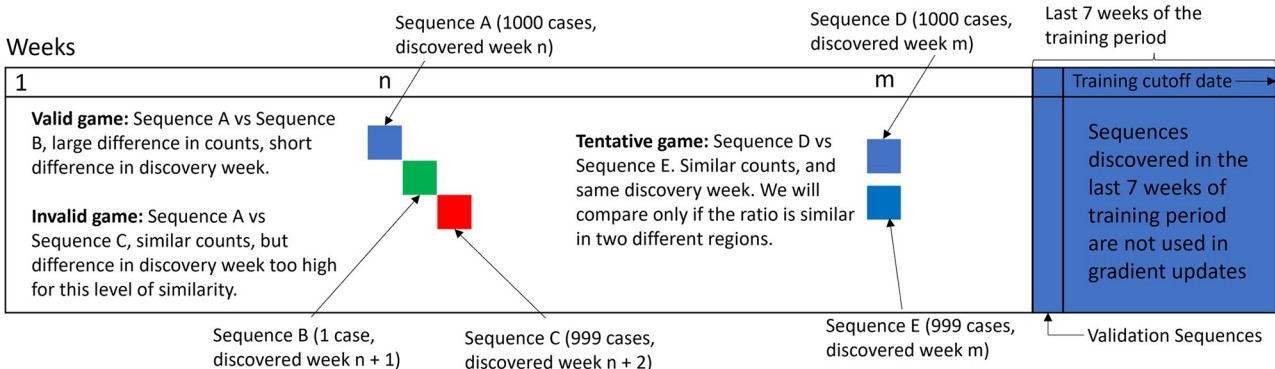

**Fig 8. Heuristics for creating *fair* games.** We allow games only among sequences first reported within a certain timeframe of each other.

rest of the world and 3) the sequences are not first reported in the last 7 weeks of the training period. If these sequences are discovered close to each other in time and they have a large difference in case counts, the competition between the sequences are accepted. Small differences in discovery dates are not deemed to upset who the winner between the sequences is in the majority of games due to the overwhelming number of cases of one sequence compared to the other. An example of this is the Sequence A vs Sequence B competition. When sequences have very similar case counts, the requirements are more stringent. In these cases, the sequences would need to be discovered in the same week. Sequence C vs Sequence A is rejected because they have almost the same case counts, but were discovered 2 weeks apart. Sequence D vs Sequence E will be considered a valid competition if the geographic conditions hold.

We apply additional heuristics so that the number of comparisons are tractable. Each sequence can be compared to a maximum of 750 sequences in competitions (random subsampling is used to cut down larger lists). If the total number of surviving competitions exceeds 1,000,000 competitions, we subsample the list to 1,000,000 cases. The case counts of two competing sequences in a pair is taken to be the cumulative case count for the first $l$ weeks where $l$ is the minimum of the number of weeks of the two sequences' lifetimes within the training period.

We use this data to train the reward model (Fig 9). During training, we fetch paired sequences from the competition draw. Lets say we selected Seq A, and Seq B. The reward model predicts an infectiousness potential for Seq A, and an infectiousness potential for Seq B, which are denoted as values $\gamma_A$, $\gamma_B$ respectively. The probability distribution of the outcome is predicted as $P(Seq\ A\ wins) = \frac{\exp(\gamma_A)}{\exp(\gamma_A) + \exp(\gamma_B)}$. This predicted distribution is trained using the "actual" distribution which is $P(Seq\ A\ wins) = \frac{N_A}{N_A + N_B}$ by minimizing the KL-divergence between the two. Note that this causes the reward model to learn to predict higher values for $\gamma_A$ if Seq A is the winner in majority of the games, hence learning to predict a potential for each sequence that increases with the infectiousness potential of the sequence.

The reward model uses the same architecture as the SDA model, except that the PM head is replaced with a reward head. A PM head produces as many outputs as there are characters in our amino acid alphabet, whereas the reward head produces a single scalar output. The rest of the layers in the reward model are initialized from the SDA model's parameter values. Also, the reward head output is only valid after the complete sequence has been passed through the model.

The reward model was trained with a learning rate of $1 \times 10^{-5}$ over 1 epoch. Learning rate warmup was instituted for the first 1024 steps, and then linearly decayed. Weight decay of $10^{-2}$ was also used. A total of four checkpoints were kept during training. For selecting the checkpoint for downstream use, we used sequences discovered in the 7th week from the end of the training period (note that these sequences were not used in training; see Fig 8). Among these, sequences having an occurrence count of over 50 in the last 7 weeks of the training period were labeled 1 and the remaining sequences were labeled 0. The sequences were each scored by the checkpoints, and the area under the receiver operating characteristic (AUROC) curve, a popular metric for model selection for binary classifiers, was computed. The checkpoint performing the best was chosen.

While we have constructed the pairwise competition draw for the sequences through robust heuristics to account for factors other than infectiousness, we note that the approaches come with some limitations and may not capture all outside variations. For instance, confounding factors such as superspreader events, and transport hubs may skew the numbers in favor of some sequences over others through short-term bursts from external factors. As such, our approach has the following limitations: (1) some data from the most recent weeks are not used to train the reward model (though it is used to train the generative model), which can result in

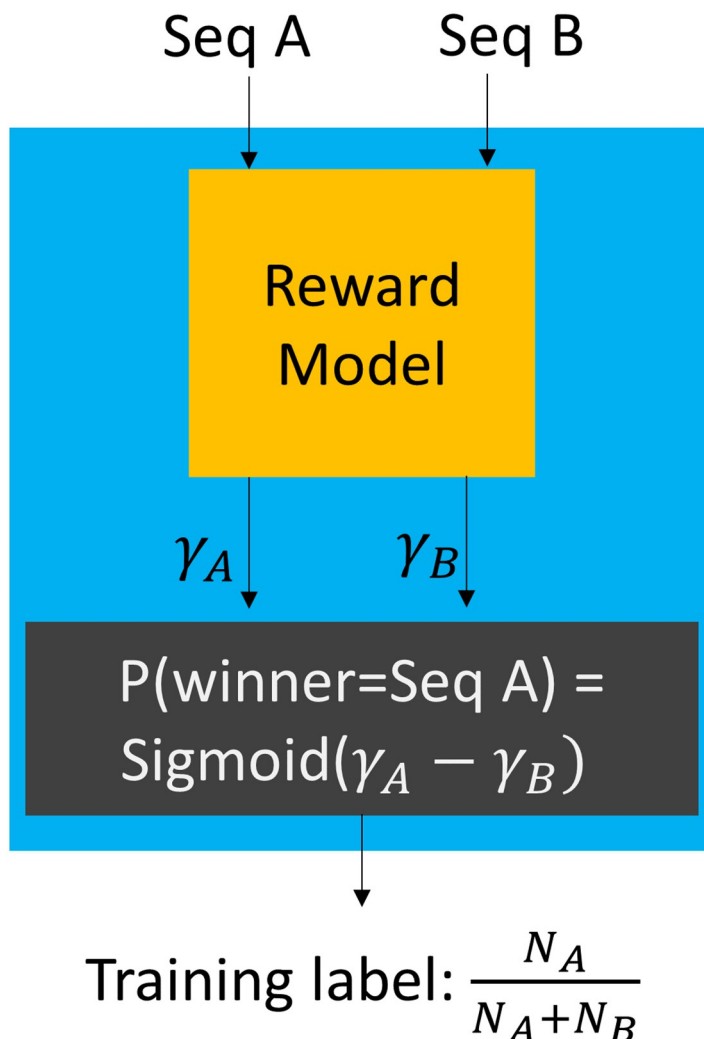

**Fig 9. Reward model training method.** The reward model processes two incoming sequences Seq A and Seq B, and produces infectiousness potentials $\sigma_A$, $\sigma_B$ for these respectively. These are converted to probabilities for competition outcomes in a differentiable manner. The probabilities are trained on the ground-truth label. The probability of Seq A winning is trained on the label value $\frac{N_A}{N_A+N_B}$ where $N_A$, $N_B$ are GISAID case counts of the two sequences within the training period.

our model not responding to certain sudden developments (2) our heuristics for ensuring fairness use sequence metadata which does not include information on specific situations in different locations or times (e.g., local events, travel patterns etc), and hence some short-term events or other locality-specific characteristics can still contaminate the data. A more thorough analysis of locations and times by integrating external knowledge such as nature of location (e.g., "transport hub"), or special events taking place in different locations (e.g., "conference" or "festival") can improve on our assumptions further. The exact form of these analyses are not clear to us presently, but they are worthy of future study.

### 4.6 PandoGen finetuning

The final step in our pipeline is to finetune the SDA model directly within its operational setting of sequence generation, using the reward model we presented in Section 4.5. To finetune

the model, in addition to the reward signal, we also use a second signal indicating whether a sequence generated by the model already exists in the list of known sequences or not.

To perform finetuning, we adapt an algorithm called Quark [25]. Quark was developed to improve an LLM's ability to generate text that conforms to certain qualities such as reduced toxicity, based on reward modeling. Quark converts floating point reward scores to discrete quanta over a pre-defined quantization scheme, and the model learns to generate sequences belonging to different reward quanta. In addition to reward modeling, using Quark, we incorporate the conditional generation framework [24], whose goal is to generate sequences of different types. We use conditional generation to incorporate the fact that data available for training can be either known in the training period, or unknown. We collapse both approaches into a single loss computation framework, as follows.

First we recollect (Section 4.2) that the SDA model generates sequences conditioned on a special beginning of sequence character. The PandoGen model, which is a finetuned version of the SDA model, generates sequences conditioned also on a second special character, which we call the conditional. Hence the PandoGen model generates sequences following the following conditional iterative sampling algorithm, where $C$ is the conditional based on which PandoGen generates sequences, and $\Phi$ represents the model's parameters.

$$x_i \sim P(x_i | X_{1:i-1}, C; \Phi) \tag{6}$$

The conditional captures two aspects: whether a sequence is known ($K$) or unknown ($U$) in the training period, and if unknown, what is the reward quantile of the sequence ($\gamma$). Hence, the full space of conditionals for the model is represented as $\mathcal{C} = \{K\} \cup \{(U, \gamma_i)\}|_{i=1}^{N_R}$, where $N_R$ is the number of reward quantiles. The conditional, $C$, essentially represents one item from the set of all conditionals. That is $C \in \mathcal{C}$.

As described in Section 1, PandoGen creates *in silico* data for training the model. For generating this data, PandoGen is conditioned on $C = (U, \gamma_H)$, where $\gamma_H$ is the highest reward quantile. Each generated sequence, $X$ is assigned a conditional, $C_X$ in two steps. $C_X = K$ if $X \in \mathcal{T}$, where $\mathcal{T}$ is the training set. If not, the reward model processes $X$ producing a reward score, which is quantized to produce $\gamma_X$; in this case, $C_X = (U, \gamma_X)$. The reward quantiles are static throughout the training process and determined based on generations from the original SDA model scored by the reward model. The generated sequences annotated with the conditionals, $(X, C_X)$ are added to a data pool, $\mathcal{D}$ of sequences (initialized using data generated from the SDA model). A fixed-size sample, $D$, is taken from this data pool, and used to perform a finetuning step. The finetuning step maximizes the following objective through mini-batch gradient descent.

$$E_{X \sim D} \sum_i \Big\{ \log P[x_i \mid X_{1:i-1}, C_X; \Phi] - \beta D_{KL}[p(x_i \mid X_{1:i-1}; \Phi_{SDA}) \| p(x_i \mid X_{1:i-1}, C_X; \Phi)] \Big\}$$

Here, $\Phi_{SDA}$ represents the parameter set of the original SDA model which is not adjusted during PandoGen finetuning, and $\Phi$ represents the parameter set of the PandoGen model, which is initialized from $\Phi_{SDA}$, and modified through the finetuning step. $D_{KL}(p\|q)$ is the KL-divergence between two distributions, $p, q$. In addition to the incorporation of conditional generation, the learning objective here is slightly different from the original Quark cost function in one more manner. The original Quark algorithm requires $D$ to be a stratified subset of $\mathcal{D}$, where sequences in each reward quantile are uniformly represented. For the default PandoGen implementation, $D$ is a simple random subset of $\mathcal{D}$.

We also implemented a looser version of stratified sampling as an option for hyperparameter tuning (explained below). In our implementation, we continue sampling from different

conditionals in $\mathcal{D}$, and stop when 1) only one conditional is left and 2) the remaining conditional has already the highest representation in $D$.

The procedure of generating sequences from the PandoGen model, scoring it using the reward model, checking historicity, andfinetuning the model using the annotated generations is carried out iteratively. Through the procedure, the PandoGen model istrained to generate sequences according to their quality and historicity (via conditioning on $C$). In the process, it learns togenerate high reward, non-historic sequences. At the same time, the loss term $\beta D_{KL}(p(x_i \mid X_{1:i-1}; \Phi_{SDA})\|p(x_i \mid X_{1:i-1}, C_X; \Phi))$ prevents the model from deviating too far from the probability distribution learnt by the original SDA model. This is important as the SDA model has a grasp of the structure of the SARS-CoV-2 Spike protein sequence, and deviating too far from the basic structure is detrimental to our goals. The training scheme is shown in Fig 2.

During training, we perform a lazy version of hyperparameter tuning to save compute resources and time. We run PandoGen finetuning steps with default hyperparameter settings for 24 epochs. The default hyperparameter case involves not using dropout, and not using stratified sampling. These settings were selected based on small-scale experiments. We use the early stopping criterion to terminate finetuning, whereby if the rewards of the generated batches (evaluated by the reward model) do not increase for three consecutive epochs, the training is halted. When training is halted, we examine the average reward for a batch generated from the best model checkpoint. We require that this reward be in the top two quantiles of rewards for samples from the SDA model prior to PandoGen finetuning. If this condition is met, we select the model for experiments. We use this criterion because the learning objective of the PandoGen model is to generate unknown sequences from the highest reward quantile. If a sample from the model achieves close to this reward on average, the model is very close to its objective, and further hyperparameter tuning may result in only marginal gains, while expending computational resources.

On the other hand if the goal is not met, we deem the training not to have succeeded, because the model is not aligned to its training objective (generating high quantile sequences) and further improvements are possible. In this case, we launch hyperparameter search. The following hyperparameter settings were explored in this case: 1) using stratified sampling, 2) no early stopping (train for all 24 epochs and select the best), and 3) train with stratified sampling and dropout. The best model checkpoint (as determined by the reward model) from these three runs is chosen for downstream experiments. Note that the model-selection method used here does not use data outside of the training period, and simply uses the reward model to determine the right model checkpoint and test convergence.

The only case where the default training hyper-parameters yielded a model which did not meet the model selection criteria was for the Delta variant experiment in Section 2.2. For this case, hyperparameter search revealed that stratified sampling and dropout provided the best results, and this model was used for further downstream experiments and analyses.

## 4.7 Temperature shaping

We summarize temperature shaping below.

Recall that the last layer of a deep autoregressive model produces the distribution of the next sequence element to be generated. The distribution is written as follows.

$$P(x_l = i \mid X_{1:l-1}) = \frac{\exp(f^i(X_{1:l-1}))}{\sum_j \exp(f^j(X_{1:l-1}))} \tag{7}$$

Here $f^i$ are Neural Network outputs. Application of temperature changes the output distribution at each time-step of the sampling process as follows.

$$P_T(x_l = i \mid X_{1:l-1}) = \frac{exp\left(\frac{f^i(X_{1:l-1})}{T}\right)}{\sum_j exp\left(\frac{f^j(X_{1:l-1})}{T}\right)} \tag{8}$$

Values of $T > 1$ flatten the output distribution and increase its entropy beyond the normal operating range, thus increasing the chance of the sampling process realizing sequences further away from the training distribution, resulting in a larger number of novel sequences in the model outputs.

## 4.8 Protein sequence generation

We used the trained models: SDA, ProGen2, Prot GPT2 enumerated, and Prot GPT2 unenumerated, to produce sequences through nucleus sampling. Batch size for generation was adjusted based on the GPU memory availability and the model size. For all cases except ProGen2, we ran five or six sampling runs at nucleus sampling values, $p \in \{0.95, 0.97, 0.99, 0.995, 0.997, 1.0\}$, in each case generating 2048 sequences. The models are also used to evaluate the log-likelihood of the sequences generated by them.

For ProGen2, we initially ran smaller scale experiments generating 256 sequences for $p \in \{0.95, 0.97, 0.99, 0.995, 0.997, 1.0\}$, with the goal of pruning cases which did not produce sufficient sequence variety. Based on this experiment, we excluded $p = 0.95$ from subsequent experiments because 88% of sequences produced from this configuration were sequences already found in the training set. For the remaining configurations, we ran ProGen2 three times in each case producing 2048 sequences. Additionally, we used the fp16 option for data generation using ProGen2 to feasibly run the experiments within our available hardware. To produce a single sample of 2048 sequences, PandoGen and Prot GPT2 were run using one V100 GPU or one A100 GPU. For ProGen2, we used 2 A100 GPUs per single sample generation run [47].

## Supporting information

**S1 File. PandoGen: Supplementary document.** Supplementary information including algorithm for sequence comparisons, details on scaling PandoGen sample size, and Performing Pango lineage reassignments.
(PDF)

## Acknowledgments

We gratefully acknowledge all data contributors, i.e., the Authors and their Originating laboratories responsible for obtaining the specimens, and their Submitting laboratories for generating the genetic sequence and metadata and sharing via the GISAID Initiative, on which this research is based.

## Disclaimer

This work does not relate to Anand Ramachandran's position at Amazon.

## Author Contributions

**Conceptualization:** Anand Ramachandran, Steven S. Lumetta, Deming Chen.

**Data curation:** Anand Ramachandran.

**Formal analysis:** Anand Ramachandran, Steven S. Lumetta.

**Investigation:** Anand Ramachandran.

**Methodology:** Anand Ramachandran, Steven S. Lumetta, Deming Chen.

**Software:** Anand Ramachandran.

**Supervision:** Steven S. Lumetta, Deming Chen.

**Validation:** Anand Ramachandran.

**Visualization:** Anand Ramachandran.

**Writing – original draft:** Anand Ramachandran.

**Writing – review & editing:** Anand Ramachandran, Steven S. Lumetta, Deming Chen.

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
