## [Decision Letter · Decision Letter 0]

26 Sep 2023

Dear Mr. Ramachandran,

Thank you very much for submitting your manuscript "PandoGen: Generating complete instances of future SARS-CoV2 sequences using Deep Learning" for consideration at PLOS Computational Biology.

As with all papers reviewed by the journal, your manuscript was reviewed by members of the editorial board and by several independent reviewers. In light of the reviews (below this email), we would like to invite the resubmission of a significantly-revised version that takes into account the reviewers' comments.

We cannot make any decision about publication until we have seen the revised manuscript and your response to the reviewers' comments. Your revised manuscript is also likely to be sent to reviewers for further evaluation.

Sincerely,

Samuel V. Scarpino

Academic Editor

PLOS Computational Biology

Arne Elofsson

Section Editor

PLOS Computational Biology

Academic editor comments

I agree with the reviewer comments that this work is potentially impactful, but also requires major revision to clarify the biological significance of the work, incorporate appropriate comparison to alternative methods, provide additional quantitative evidence for some of the more substantive claims, improve the discussion of similar/relevant past studies, and address concerns regarding the use of GISAID data. Regarding the last point, I have quite a bit of familiarity with GISAID and am convinced their Pango lineage assignments are often out-of-date or incorrect. I am also aware that downloading all the data used in this study and running your own assignments isn't an option because of limitations with GISAID. My suggestion is select a random subset of sequences used in your study up to the max allowable download, assign them with Pango lineages, and report the congruence in the appendix. I don't believe that 100% congruence here is necessary for acceptance, but the authors should provide strong justification for why we should believe your results are robust to lineage miss-assignment. I also want to stress that the reviewers are suggesting a substantial amount of work and that many of the items will require running additional computational experiments. I do not believe most of their concerns can be adequately addressed by adding caveats to the text. Regarding the code, I believe that what is provided is technically "over the bar" but also agree with R2 that more time should be invested in documenting the code and ensuring it's easy for interested users to recreate the results and utilize the code for their own studies.

Lastly, R1 added their comments as confidential to the editors, but requested that we communicate them directly to you. I have pasted them below. Please address them as you would any other reviewer comments when submitting a revision. Thank you for sending us your work and I'll hope to see a revision from you soon!

Reviewer 1 comments:

Overall, I think the authors have presented a well-structured pipeline in generating novel sequences with good predictability. The use of a reward model to predict potential infectiousness is innovative. The objective function combining a MLE for sequence prediction and a penalty term to limit deviation from the SARS-CoV2 spike protein sequence is very interpretable. And their results has shown capability to produce novel sequences that later occurred as SARS-CoV2 sequences. I believe the authors have presented sufficient results to show that PandoGen is capable of producing sequences that are not in the training set. And they have demonstrated predictability of the model both in fig2 and in fig3 with the improved PPV results when considering high rank sequences. Additionally, it is nice to see Pandogen’s PPV score improves with a stricter top-n sequence ranking threshold as it shows efficacy of the ranking measures.

In terms of presentation of the methods and results, I think readability and accessibility could be improved.

For example:

1. Since the performance evaluation metrics employed in this work are proposed by the author, it would be helpful if the evaluation measures were clearly defined and stated.

2. Another thing is the reward system is mentioned early in the manuscript but only explained further in section 5.

3. I would also like to suggest mentioning of the length of the sequences used for the analysis since it impacts the computation costs a lot.

4. I am curious about the number of sequences generated to obtain the top-ranking sequences, especially considering the improvement from around 30% PPV in Figure 2 to approximately 20%-70% PPV in Figure 3.

5. Moreover, since the methods used for comparison produce significantly fewer novel sequences, I find the evidence supporting 'better performance in predictability' not very strong. Please discuss why ~30% predictability (PPV) should be considered a "good result".

6. Relatedly, in figure 2, the methods used for comparison to the proposed PandoGen rarely produce novel sequences. I think that explains why Prot GPT2 enumerated is showing a big variance in the PPV plot (due to a very small denominator). In my opinion, if all methods are producing a lot of novel sequences, but PandoGen is dominating PPV then it would be more convincing to claim PandoGen performs better in terms of novel sequence prediction.

Reviewer's Responses to Questions

**Comments to the Authors:**

Reviewer #1: See above

Reviewer #2: The authors of this paper have achieved an impressive feat of training an AI to predict future sequences of the SARS-CoV-2 virus. The work appears to be mostly methodologically sound. However, the current version of the manuscript is heavy going, but key methodological details and results are left lacking, as is a robust discussion of how this work relates to existing AI models already published in some landmark studies. Whilst the authors mention other works, their discussion of these and what is novel about their work is lacking, it’s hard to evidence originality with the manuscript in it’s current form.

Furthermore, they don’t robustly compare their work to these other methods, focusing on a single competing methodology only. There are also issues of the biggest claims made by the work being presented in rather vague terms, with the details buried within supplemental information or a terse methods section, which feels off and like an effort to evade scrutiny, clearly more work needs to be done to prove that that methods are robust and I highlight a few key areas where more discussion is needed of shortcommings, and I’m especially concerned about the suitability of the PANGO lineage classifications used from GISAID. The work is rather lacking in biologically meaningful conclusions, feeling rather like an incremental improvement on prior works. I also can’t find any mention of code availability, which is troubling. I believe that with a reworked manuscript, this work could potentially be acceptable for publication; I outline the issues I would like to see addressed below:

1) The canonical name for the virus that causes COVID-19 is SARS-CoV-2 and not “SARS-CoV2”, as used throughout the manuscript. This should be corrected to reflect the current literature.

2) The authors use a discriminative model to train the AI for an “infectiousness potential”, using pairwise competitions between sequences which are temporally close within the pandemic. The authors do state that there are issues with inferring this from data within GISAID alone, and I believe the authors have approximated the first order of this problem successfully using occurrence data alone, and they appear to use robust heuristics for the comparisons. However, there are issues here in that, potentially, one sequence is conferred an advantage over another due to factors not captured within their model. Such a viral sequence first occurring at a superspreader event or within a transport hub of a densely populated city would have a much more significant “founder effect” than a new viral sequence first occurring in a remote location. While it is potentially challenging to model this effect, it would be good for the authors to more thoroughly acknowledge the limitations in their works and that going off the frequency of sequence observation alone is not a perfect solution. I would like a more detailed discussion of the founder effect within the main body of the manuscript and how this confounds their “infectiousness” potential.

3) In line 165, the authors state that PandoGen forecasts many of these variants ahead of time; they need to give more details here for a claim like this.

4) Line 312 specifically this statement: “sequences with 10 or more cases constitute less than 10% of all sequences.” this whole concept needs better explanation: do they mean 10 or more cases in their predicted sequences? The authors’ concept of what constitutes a “salient” sequence needs a better description so as not to confuse the reader, especially since this is a key concept of the manuscript.

5) On line 341, the authors state: “PandoGen predicts a significant proportion of all novel lineages reported for many weeks after training.” This and the following line is too vague a claim. They need to be more specific when discussing the performance of their methods; specifically, which time point do they mean here?

6) Line 353, the comment “We generate and analyze samples to see whether important pandemic variants such as Delta and Omicron sublineages can be consistently generated by PandoGen” about the ability to generate the sequences of Delta and Omicron requires much more detail. This is potentially a significant finding, but doesn't seem fully supported.

7) In lines 359-362, the authors compare their method to that of Prot GPT 2. However, all details are buried within the supplemental; a summary of these findings must be present within the main body of the manuscript. Equally, should a comparison to other methods not be made at this point too?

8) Around line 386, the authors discuss using data within GISAID, specifically the Pango lineage, as training data. However, can we depend on GISAID assignments here? Would it not be better to download and fully reassign lineages to all GISAID sequences? Because until PANGO defines a lineage, it can’t be assigned by pangolin? Thus, the validity of the whole study here is predicated on the assumption that GISAID respectively updates its lineage assignments, and I can’t find any documentation stating they do! Potentially, they may have trained their models naively on sequences that GISAID missed, as the PANGO definitions for Delta and Omicron have not yet been released or updated such that GISAID could not classify these early sequences as such. This point needs to be cleared up as the whole study is based on an assumption of arcuate lineage assignment within GISAID, is this re-visited and updated by GISAID, or is it only performed once at sequence submission time?

9) The tables need legends describing the content of the columns.

10) The related works section seems to mention many key landmark studies which also use AI-based methods to predict or assess sequences of the virus published in Science and Cell in passing without the usual reference to what the authors do that is different or how their presented work is novel or goes beyond these studies. Rather, a commentary is made on the methodological underpinnings of these works but not how this work supersedes or supplements prior work within the field. This section must be substantially reworked to show what is novel about the presented work and how it is differentiated from prior works in this area.

11) Code availability: there appears to be no discussion of the code used in this study within the main body of the manuscript. For reproducibility, the code used to generate and train the models and assess the performance relative to other works needs to be placed within GitHub or a similar repository under an open-source license. This is an essential pre-request for publication. A URL is given in the tables that pre-pend the submitted manuscript however that GitHub repo has very little documentation, as such I don't feel the methods are reproducible without better documented code.

**Have the authors made all data and (if applicable) computational code underlying the findings in their manuscript fully available?**

Reviewer #1: None

Reviewer #2: **No: **I can't find any reference to code availability within the main body of the manuscript. A URL is given in the tables that pre-pend the submitted manuscript however that GitHub repo has very little documentation, as such I don't feel the methods are reproducible without better documented code.

PLOS authors have the option to publish the peer review history of their article (what does this mean?). If published, this will include your full peer review and any attached files.

Reviewer #1: No

Reviewer #2: No
---

## [Decision Letter · Decision Letter 1]

27 Dec 2023

Dear Mr. Ramachandran,

We are pleased to inform you that your manuscript 'PandoGen: Generating complete instances of future SARS-CoV-2 sequences using Deep Learning' has been provisionally accepted for publication in PLOS Computational Biology.

Best regards,

Samuel V. Scarpino

Academic Editor

PLOS Computational Biology

Arne Elofsson

Section Editor

PLOS Computational Biology

Reviewer's Responses to Questions

**Comments to the Authors:**

Reviewer #2: I'm satisfied that my comments on this manuscript have been addressed, and that the reworked manuscript is substantially improved, with sufficient details present in sections that we previously lacking, a better comparison to prior work, and issues of code availability and PANGO lineages addressed. I feel that this revision is now acceptable for publication.

**Have the authors made all data and (if applicable) computational code underlying the findings in their manuscript fully available?**

Reviewer #2: Yes

PLOS authors have the option to publish the peer review history of their article (what does this mean?). If published, this will include your full peer review and any attached files.

Reviewer #2: No

---

## [Editor Report · Acceptance letter]

17 Jan 2024

PCOMPBIOL-D-23-01345R1 

PandoGen: Generating complete instances of future SARS-CoV-2 sequences using Deep Learning

Dear Dr Ramachandran,

I am pleased to inform you that your manuscript has been formally accepted for publication in PLOS Computational Biology. Your manuscript is now with our production department and you will be notified of the publication date in due course.

With kind regards,

Judit Kozma
